# DETAILMASTER: Can Your Text-to-Image Model Handle Long Prompts?

Qirui Jiao [* 1]  Daoyuan Chen [* 2]  Yilun Huang [2]  Xika Lin [3]  Ying Shen [† 1 4]  Yaliang Li [2]

## Abstract

While recent Text-to-Image (T2I) models show impressive capabilities in synthesizing images from brief descriptions, they struggle with the long, detailed prompts required for professional applications. We present **DETAILMASTER**, a comprehensive benchmark for evaluating T2I capabilities on long prompts with complex compositional requirements, accompanied by an automated data construction pipeline and an evaluation workflow. Comprising expert-validated prompts averaging 284.89 tokens, our benchmark introduces four critical evaluation dimensions: Character Attributes, Structured Character Locations, Multi-Dimensional Scene Attributes, and Spatial/Interactive Relationships. Evaluations on various general-purpose and long-prompt-optimized models reveal critical performance limitations, showing that weak encoders struggle to preserve syntactic dependencies within prompts and diffusion models suffer from attribute leakage under detail-intensive conditions. Through a controlled ablation study under varying constraints, we further show that high-fidelity generation requires a synergistic combination of expanded prompt limits and long-prompt training. We open-source our dataset and code to foster progress in long-prompt-driven T2I generation.

## 1. Introduction

While Text-to-Image (T2I) generation has rapidly progressed in perceptual quality and semantic alignment (Betker et al., 2023; Labs, 2024), models may struggle to interpret long, detailed prompts. Nevertheless, many real-world workflows require long prompts that specify mul-

tiple entities, attributes, locations, and interactions (Cao et al., 2025; Qin et al., 2025b; Wu et al., 2024a). These long prompts are not just "short prompts with more adjectives." They introduce **high constraint density** (i.e., many requirements competing for limited generative capacity), **structured compositionality** (i.e., entities with nested modifiers and clause-level dependencies), and **binding under load** (i.e., attributes must be attached to the correct entities, not merely present).

The limitations in faithfully following long prompts can be attributed to four constraints (Liu et al., 2025; Hu et al., 2024): 1) **Training data bias.** Classical models are trained on short prompts (e.g., COCO (Lin et al., 2014) averaging 10.5 tokens, CC12M (Changpinyo et al., 2021) averaging 20.2 tokens), which reinforces a preference for concise inputs. 2) **Structural comprehension deficiency.** Many text encoders fail to adequately parse hierarchical descriptions involving multiple objects, attributes, and their spatial relationships. 3) **Detail overload.** When prompts contain excessive descriptive details on a single entity, models tend to omit or distort key details. 4) **Token length constraints.** Many text encoders impose strict upper bounds on input tokens (e.g., CLIP's 77-token limit (Radford et al., 2021)).

While existing evaluations have examined T2I models across multiple dimensions, including image-text alignment, attribute binding, and human preference, they predominantly rely on short prompts. Only a few benchmarks consider long prompts (Liu et al., 2025; Hu et al., 2024), but they remain shorter and simpler than professional instructions, and often rely on coarse metrics (e.g., CLIPScore (Hessel et al., 2021) or limited binary checks), which provide weak diagnostic capability for fine-grained compositional errors.

To rigorously evaluate T2I models' prompt-following abilities on long prompts, we introduce **DETAILMASTER**, a novel benchmark along with an automated data construction pipeline and a robust evaluation workflow. This benchmark assesses prompt adherence across four critical dimensions, specifically designed to target two challenges: structural comprehension deficiency (via **Character Locations** and **Spatial/Interactive Relationships**) and detail overload (via **Character Attributes** and **Scene Attributes**). The prompts in our benchmark are derived from existing detailed captions (Onoe et al., 2024; Pont-Tuset et al., 2020), which we

*Equal contribution. †Corresponding author. [1]Sun Yat-Sen University [2]Alibaba Group [3]Worcester Polytechnic Institute [4]Guangdong Provincial Key Laboratory of Fire Science and Intelligent Emergency Technology. Correspondence to: Ying Shen <sheny76@mail.sysu.edu.cn>.

*Proceedings of the $43^{rd}$ International Conference on Machine Learning*, Seoul, South Korea. PMLR 306, 2026. Copyright 2026 by the author(s).

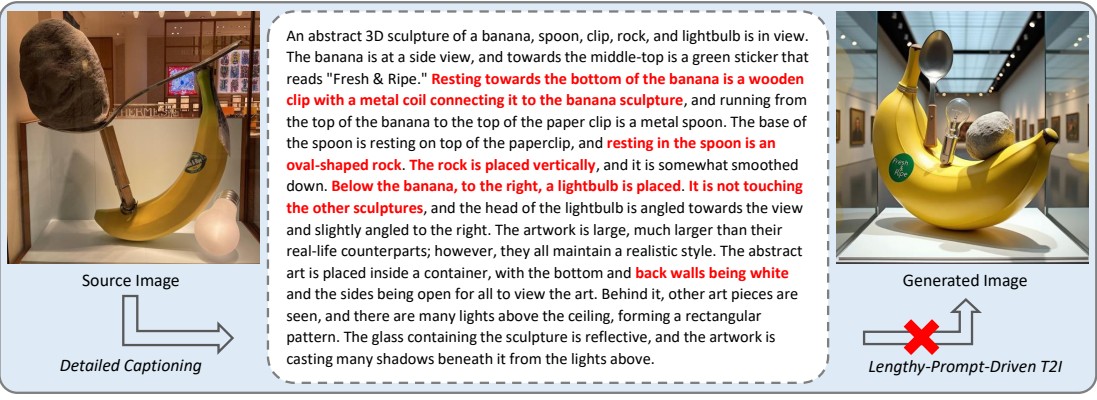

**An abstract 3D sculpture of a banana, spoon, clip, rock, and lightbulb is in view. The banana is at a side view, and towards the middle-top is a green sticker that reads "Fresh & Ripe." Resting towards the bottom of the banana is a wooden clip with a metal coil connecting it to the banana sculpture, and running from the top of the banana to the top of the paper clip is a metal spoon. The base of the spoon is resting on top of the paperclip, and resting in the spoon is an oval-shaped rock. The rock is placed vertically, and it is somewhat smoothed down. Below the banana, to the right, a lightbulb is placed. It is not touching the other sculptures, and the head of the lightbulb is angled towards the view and slightly angled to the right. The artwork is large, much larger than their real-life counterparts; however, they all maintain a realistic style. The abstract art is placed inside a container, with the bottom and back walls being white and the sides being open for all to view the art. Behind it, other art pieces are seen, and there are many lights above the ceiling, forming a rectangular pattern. The glass containing the sculpture is reflective, and the artwork is casting many shadows beneath it from the lights above.**

Source Image

Detailed Captioning

Generated Image

Lengthy-Prompt-Driven T2I

*Figure 1.* Text-to-image errors in long prompt scenario (with FLUX.1-dev). Real source image (left), detailed caption/prompt (middle), and generated image (right). Red text indicates failure points.

further refine by augmenting missing details, resulting in an average token length of 284.89. To ensure high quality, human experts are engaged to evaluate our benchmark, confirming its high standard. For the evaluation workflow, we develop specialized assessment mechanisms for each category of attributes, enabling systematic evaluation of models' compositional T2I abilities in long-prompt scenarios.

Using our DETAILMASTER benchmark, we conduct a comprehensive evaluation of 19 representative text-to-image models. To capture the rapid evolution of the field, our evaluation spans five distinct paradigms: (1) Classical & Flow-based Diffusion models, (2) Long-prompt optimized models (based on SD1.5 (Rombach et al., 2022) or SD-XL (Podell et al., 2024)), (3) Models equipped with advanced LLM or MLLM encoders (e.g., Qwen-Image (Wu et al., 2025b)), (4) Emerging autoregressive and unified architectures (e.g., the Janus family (Wu et al., 2025a; Ma et al., 2025; Chen et al., 2025b), BAGEL (Deng et al., 2025)), and (5) Proprietary State-of-the-Art models (e.g., GPT Image-1 (OpenAI, 2025), Gemini 2.0 Flash (Kampf & Brichtova, 2025)). While our analysis validates enhanced competencies in recent models over older baselines, we identify a consistent struggle with fine-grained compositional elements.

Additionally, a statistical analysis indicates a clear difficulty hierarchy, with Character Locations and Person Attributes proving the most challenging. Furthermore, through a controlled ablation study with a modified SD-XL architecture, we demonstrate that performance gains are not only a function of longer context windows, but also depend on training with detailed, longer prompts. And there is a consistent degradation in prompt adherence as prompt length increases, highlighting challenges in handling long prompts.

In summary, our contributions are: 1) We present an automated data construction pipeline and generate a long-prompt T2I benchmark with 4,116 prompts (avg. 284.89 tokens) with structured constraint annotations. 2) We propose a fine-grained evaluation workflow that scores prompt-following across four compositional dimensions, with cross-evaluator robustness checks. 3) We conduct extensive evaluations and ablation studies, yielding several insights: current models show a degradation in adherence as prompt length increases; progress in long-prompt adherence is driven less by expanded context windows and more by factors such as long-prompt training and iterative decomposition. 4) We open-source all data and code (*GitHub Link*) to advance research in T2I generation for long, detail-rich prompts.

**Conflict of Interest Disclosure:** The authors Yaliang Li, Daoyuan Chen, and Yilun Huang are employed by Alibaba Group, the developer of Qwen-Image, which was evaluated in this paper. To ensure transparency, all code and datasets are publicly available, and our results are based strictly on reproducible, quantitative metrics without bias.

## 2. Related Works

**Text-to-image models.** Text-to-image (T2I) generation aims to synthesize semantically aligned images conditioned on textual descriptions. Initial endeavors use Generative Adversarial Networks (GANs) (Goodfellow et al., 2014), employing a generator-discriminator framework. Subsequent advancements introduce autoregressive models that treat image generation as a sequence prediction task, including DALL·E (Ramesh et al., 2021) and Parti (Yu et al., 2022). More recently, diffusion-based models (Ho et al., 2020) have emerged as a powerful paradigm. These models, including DALL·E 2 (Ramesh et al., 2022), Imagen (Saharia et al., 2022), and Stable Diffusion (Rombach et al., 2022), operate by iteratively denoising to produce coherent images. Beyond this, flow-based models such as Stable Diffusion 3.5 (Esser et al., 2024) and FLUX (Labs, 2024) use flow matching (Lipman et al., 2022) to learn transformations from noise to images, reducing sampling complexity and improving efficiency. Furthermore, the proprietary models Gemini

*Table 1.* Comparison of data composition between **DETAILMASTER** and other benchmarks. Statistical analysis is in Section 5.1.

| Benchmark | # Prompts | # Nouns | Character | | # Scene Attributes | # Enitity Relationships | Avg. Tokens |
| | | | # Attributes | # Locations | | | |
| --- | --- | --- | --- | --- | --- | --- | --- |
| HRS-Comp (Bakr et al., 2023) | 3004 | 620 | 1000 | N/A | N/A | 2000 | 16.43 |
| T2I-CompBench (Huang et al., 2023) | 6000 | 2316 | 4000 | N/A | N/A | 2000 | 12.65 |
| DensePrompts (Liu et al., 2025) | 7061 | 4913 | N/A | N/A | N/A | N/A | 100.04 |
| DPG-Bench (Hu et al., 2024) | 1065 | 4286 | 5020 | N/A | 329 | 2593 | 83.91 |
| **DETAILMASTER (Ours)** | 4116 | 5165 | 37165 | 6910 | 12330 | 18526 | 284.89 |

2.0 Flash (Kampf & Brichtova, 2025) and GPT Image-1 (OpenAI, 2025) also show SOTA level performance.

**Benchmarks for T2I generation.** Early evaluations for T2I generation primarily use datasets such as CUB Birds (Wah et al., 2011) and Oxford Flowers (Nilsback & Zisserman, 2008), which have limited diversity and simple prompts. Recently, benchmarks such as DrawBench (Saharia et al., 2022), HRS-Bench (Bakr et al., 2023) and T2I-CompBench (Huang et al., 2023) introduce prompts aimed at compositional generation, including object presence, attribute binding, and spatial relationships. Further advancing this direction, ConceptMix (Wu et al., 2024b) evaluates compositional limits by automatically generating prompts with a controllable number of combined concepts. To measure every detail within the prompts, benchmarks such as TIFA (Hu et al., 2023) and Gecko (Wiles et al., 2025) decompose prompts into elemental components and generate corresponding questions to assess model fidelity. Moreover, benchmarks such as GenAI-Bench (Li et al., 2024a), RichHF18K (Liang et al., 2024) and HPS v2 (Wu et al., 2023b) train preference-aligned evaluators that reflect human judgment. However, these benchmarks mainly focus on short prompts, neglecting the challenges of longer, complex inputs that are prone to issues like attribute misalignment.

**T2I models for long prompt.** Classical models such as SD1.5 (Rombach et al., 2022) and SD-XL (Podell et al., 2024) are constrained by the 77-token limit of their CLIP text encoder, hindering performance on long prompts as they have to be trained and used on short prompts. Beyond this, some studies attempt to extend the prompt limit. Representatives such as Imagen (Saharia et al., 2022), DeepFloyd IF (StabilityAI, 2023), Stable Diffusion 3.5 (Esser et al., 2024) and FLUX (Labs, 2024) adopt T5 (Raffel et al., 2020), extending the limit to up to 512. However, these models are trained on short prompts, whereas the subsequent models are trained on or designed with optimizations for long prompts. For instance, ParaDiffusion (Wu et al., 2025c) uses Llama V2 (Touvron et al., 2023) as its text encoder and trains on long prompts. LLM4GEN (Liu et al., 2025) proposes a specialized loss to penalize attribute mismatch. ELLA (Hu et al., 2024) incorporates a trainable mapper after its encoder to improve information extraction. LongAlign (Liu et al., 2024) decomposes long prompts into individual sentences for separate encoding, while also enabling preference opti-

mization for long-prompt alignment. LLM Blueprint (Gani et al., 2024) extracts object details from long prompts using an LLM, then employs layout-to-image generation and an iterative refinement scheme. However, due to the scarcity of benchmarks for long-prompt-driven image generation, these methods lack fair comparison.

**Benchmarks for long-prompt-driven T2I generation.** Current benchmarks for long-prompt-driven image generation exhibit several limitations. Existing works (e.g., DensePrompts (Liu et al., 2025), DPG-Bench (Hu et al., 2024), LongBench-T2I (Zhou et al., 2025), T2I-CoReBench (Li et al., 2025), OneIG-Bench (Chang et al., 2026)) explore complex instructions but fall short in three key areas: 1) Prompt Source & Logic: Many benchmarks, such as LongBench-T2I and T2I-CoReBench, rely heavily on LLM-generated or entirely synthetic prompts. While convenient for scaling, these artificially constructed prompts often risk violating real-world physical logic. 2) Prompt Density: Existing datasets frequently fail to match true real-world instructional complexity. For instance, only 25% of OneIG-Bench prompts exceed 60 tokens. DensePrompts and DPG-Bench similarly remain much shorter than professional use cases. 3) Evaluation Granularity: Previous works predominantly rely on coarse metrics like CLIPScore (Hessel et al., 2021), overall semantic alignment, or basic VQA checklists, which lack the diagnostic capability for fine-grained compositional errors.

To address these limitations, we introduce **DETAILMASTER**. Grounded in real image-caption pairs to ensure visual realism and structural logic, our benchmark features dense prompts (averaging 284.89 tokens) and employs a hierarchical, multi-model verification pipeline to strictly isolate and evaluate individual attributes, locations, and relationships. As shown in Table 1, our benchmark advances prior work through extended prompts, enlarged attribute sets, and more comprehensive evaluation objectives.

## 3. The DETAILMASTER Benchmark

### 3.1. Dataset Construction

#### 3.1.1. OVERVIEW

The primary objective of our dataset construction is to simulate "detail-overload" scenarios. By purposefully creating

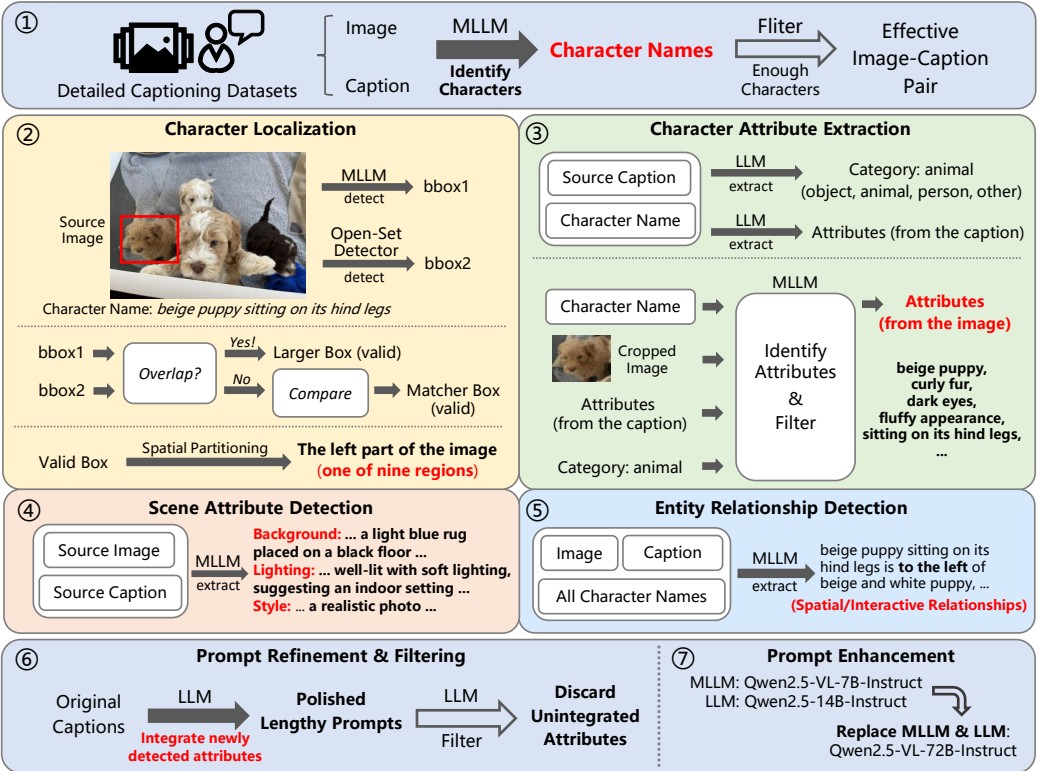

*Figure 2.* Overview diagram of the data construction process for the **DETAILMASTER** benchmark.

high constraint density, multi-entity relationships, and strict positional requirements, this design explicitly serves as a proxy for professional-grade prompt adherence rather than standard user behavior.To better simulate the challenges in long-prompt scenarios, we require prompts with sufficient length and rich details. Hence, we leverage human-annotated captioning datasets, where each sample comprises an image paired with fine-grained textual descriptions, including DOCCI (Onoe et al., 2024) and Localized Narratives (Flickr30k (Young et al., 2014) and COCO subsets) (Pont-Tuset et al., 2020). While these datasets already provide precise object descriptions with an average token length of 68.4, their coverage remains incomplete. To obtain better prompts, we design a fine-grained attribute extraction pipeline that systematically captures four categories of features: 1) **Character Attributes**, where we categorize main characters (objects, animals, persons, and others) and analyze them via class-specific attribute extraction; 2) **Structured Character Locations**, where we employ a nine grid-based spatial partitioning scheme to get position information; 3) **Multi-Dimensional Scene Attributes**, decomposing scenes into background, lighting, and style elements; and 4) **Spatial/Interactive Relationships**, annotating character pairs with geometric (e.g., "dog behind chair") and dynamic interactions (e.g., "holding"). These extracted attributes are then integrated into the original captions using

LLM-based expansion, yielding detailed prompts with an average token length of 284.89. Notably, since the annotations in evaluation match the final prompts rather than strictly following the source images, including more details in the final prompts does not introduce evaluation errors. The overview is shown in Figure 2.

### 3.1.2. ATTRIBUTE EXTRACTION PIPELINE

**Main character identification.** Initially, we provide an MLLM with both the image and the caption to identify the characters present within each sample. We then apply filtering to retain only instances containing more than four main characters, ensuring as many high-quality features as possible in the final prompt (see Figure 2 ①).

**Character localization.** To localize a character, we generate two bounding box proposals using an MLLM and an open-set detector called YOLOE-11L (Wang et al., 2025). We then validate these proposals through a two-stage process. First, if the boxes have an Intersection over Union (IoU) of at least 0.7, we select the larger one. Otherwise, we employ BLIP (Li et al., 2022b) to score each candidate area against the character's name and then retain the higher-scoring box unless both scores fall below 0.4 (see Figure 2 ②). Finally, the validated bounding boxes are converted into structured spatial descriptions using the nine-region

partitioning scheme detailed in Appendix B.

**Character attribute extraction.** For character attributes, we implement a hierarchical pipeline. First, an LLM performs text-based extraction by classifying characters (object, animal, person, other) and using category-specific prompts to derive attributes from captions (e.g., material for objects, clothing for persons). Subsequently, a multimodal refinement stage enhances attribute completeness and simulate detail overload. For characters with location information, we crop corresponding sub-images and ask an MLLM to supplement the text-derived attribute lists with visually-grounded features. Finally, we use the MLLM to discard mismatched attributes (see Figure 2 ③).

**Scene attribute extraction.** For scene attributes, we employ an MLLM to jointly analyze the source image and the caption, identifying its background composition, lighting conditions, and stylistic elements (see Figure 2 ④).

**Entity relationship detection.** For entity relationships, we input the source image, the original caption, and all detected characters of each sample into the MLLM. The model then performs relationship parsing to extract all discernible spatial and interactive relationships between all characters (see Figure 2 ⑤).

### 3.1.3. PROMPT REFINEMENT & ENHANCEMENT

**Prompt refinement and filtering.** Following the attribute extraction, we implement a prompt refinement pipeline to transform the original captions into detail-rich long prompts. For each sample, we enumerate all identified main characters and employ an LLM to incorporate all their attributes and locations into the original caption. Subsequently, the same LLM integrates the sample's scene attributes and spatial/interactive relationships to produce the refined prompt.

Beyond the prompt refinement, the extracted attributes also form the basis of our evaluation metrics. To ensure metric accuracy, we implement an attribute validation mechanism where an LLM verifies the presence of each attribute in the refined prompt, filtering out any unincorporated attributes. In addition, samples with less than 4 valid character-level attributes are filtered out (see Figure 2 ⑥).

**Prompt enhancement.** Our data construction process is carried out in two rounds. In the first round, we use image data and detailed captions from DOCCI and Localized Narratives as sources, and deploy Qwen2.5-14B-Instruct (Team, 2024) and Qwen2.5-VL-7B-Instruct (Team, 2025) for initial data synthesis. This round yields 4,565 detail-rich prompts, with an average token length of 237.53. In the second round, we use these 4,565 samples as the original data and rerun the synthesis pipeline, employing Qwen2.5-VL-72B-Instruct as both the LLM and MLLM. This reprocessing step results in an improved dataset containing 4,116 refined prompts with

an average token length of 284.89 (see in Figure 2 ⑦).

### 3.2. Evaluation Workflow

#### 3.2.1. OVERVIEW

Evaluating compositional T2I generation proves inherently difficult, as it demands fine-grained cross-modal understanding between textual prompts and generated images. We introduce a robust, multi-stage evaluation pipeline that systematically assesses attribute fidelity across four categories.

#### 3.2.2. EVALUATION METRICS

Our evaluation framework is built upon the four categories of attributes that we previously extract. For the **"Character Attributes"** metric, we calculate it as the ratio of correctly rendered attributes to the total number of attributes for each character category (i.e., object, animal, and person). The **"Character Locations"** metric assesses accuracy by computing the ratio of characters positioned correctly to the total number of annotated characters. For **"Entity Relationships"**, we calculate the percentage of correctly rendered relationships among all those described in the prompt. These three metrics quantify the model's performance on attribute-level details, specifically targeting mismatches and omissions. For the **"Scene Attributes"** metric, we compute accuracy for background, lighting, and style, evaluating overall image fidelity in long prompt scenarios.

Our main evaluations are conducted on the full version of our **DETAILMASTER**, comprising 4,116 prompts. Additionally, to facilitate rapid evaluation, we design a mini-benchmark comprising 800 detail-rich prompts (detailed in Appendix N). Moreover, we also provide an option that employs a smaller evaluator to accommodate researchers with limited GPU VRAM (detailed in Appendix K).

#### 3.2.3. EVALUATION PIPELINE

We introduce a multi-step evaluation pipeline to systematically assess the compositional generation capabilities of T2I models. First, we instruct the T2I models with prompts from our **DETAILMASTER** benchmark to generate corresponding image sets. Second, we localize the characters present in generated images. Subsequently, we conduct a rigorous quantitative evaluation across the four metrics, deriving accuracy rates for each evaluated model. Further details of the evaluation pipeline are provided in Appendix F.

## 4. Experiments

### 4.1. Experimental Setup

We evaluate a diverse suite of 19 models. As categorized in Table 2, these models span five distinct paradigms rep-

*Table 2.* Results on the **DETAILMASTER** Benchmark. Values represent accuracy percentages.

| Model | Backbone | Character Attributes | | | Character Locations | Scene Attributes | | | Entity Relationships |
|---|---|---|---|---|---|---|---|---|---|
| | | Object | Animal | Person | | Background | Light | Style | |
| (A) Classical & Flow-based Diffusion Models | | | | | | | | | |
| SD1.5 (Rombach et al., 2022) | - | 20.79 | 27.69 | 13.89 | 8.68 | 22.02 | 64.52 | 80.90 | 5.88 |
| SD-XL (Podell et al., 2024) | - | 24.41 | 29.54 | 16.73 | 10.95 | 27.52 | 68.42 | 68.87 | 9.83 |
| DeepFloyd IF (StabilityAI, 2023) | - | 31.01 | 37.47 | 25.61 | 14.28 | 26.26 | 67.87 | 86.49 | 11.95 |
| SD3.5 Large (Esser et al., 2024) | - | 48.56 | 46.20 | 32.95 | 33.62 | 89.61 | 90.33 | 95.69 | 40.03 |
| FLUX.1-dev (Labs, 2024) | - | 51.47 | 45.83 | 34.91 | 41.57 | 95.77 | 97.05 | 94.81 | 47.49 |
| (B) Long-Prompt Optimized Models (Based on SD1.5/SD-XL) | | | | | | | | | |
| LLM4GEN (Liu et al., 2025) | SD1.5 | 21.75 | 29.14 | 17.20 | 9.22 | 25.60 | 65.71 | 50.11 | 7.44 |
| LLM Blueprint (Gani et al., 2024) | SD1.5 | 21.41 | 27.01 | 13.91 | 18.44 | 50.89 | 77.57 | 64.24 | 11.60 |
| ELLA (Hu et al., 2024) | SD1.5 | 34.50 | 35.14 | 22.01 | 15.92 | 47.12 | 74.28 | 40.14 | 16.64 |
| LongAlign (Liu et al., 2024) | SD1.5 | 32.95 | 34.60 | 15.35 | 14.89 | 77.03 | 87.70 | 72.27 | 19.17 |
| ParaDiffusion (Wu et al., 2025c) | SDXL | 35.25 | 33.28 | 22.29 | 20.32 | 83.65 | 92.21 | 67.45 | 25.50 |
| (C) Models with an LLM/MLLM Encoder | | | | | | | | | |
| SANA (Xie et al., 2024) | - | 40.79 | 38.88 | 24.74 | 22.91 | 89.80 | 94.51 | 73.34 | 29.56 |
| Qwen-Image (Wu et al., 2025b) | - | 51.01 | 49.11 | 39.30 | 46.98 | 98.35 | 98.98 | 96.08 | 60.04 |
| (D) Autoregressive Models & Unified Models | | | | | | | | | |
| Janus-1.3B (Wu et al., 2025a) | - | 35.86 | 32.15 | 22.31 | 21.40 | 87.59 | 95.67 | 83.53 | 25.48 |
| JanusFlow-1.3B (Ma et al., 2025) | - | 31.93 | 37.54 | 16.50 | 17.30 | 81.24 | 94.80 | 92.99 | 16.77 |
| Janus-Pro-1B (Chen et al., 2025b) | - | 40.55 | 41.44 | 26.10 | 26.66 | 92.39 | 96.42 | 88.78 | 32.30 |
| Lumina-Image-2.0 (2B) (Qin et al., 2025a) | - | 43.74 | 43.79 | 29.20 | 34.28 | 86.86 | 93.47 | 93.81 | 39.68 |
| BAGEL-7B-MoT (Deng et al., 2025) | - | 47.04 | 46.35 | 33.80 | 39.59 | 97.41 | 97.99 | 94.68 | 49.43 |
| (E) Proprietary SOTA Models | | | | | | | | | |
| Gemini 2.0 Flash (Kampf & Brichtova, 2025) | - | 55.44 | 47.84 | 34.23 | 44.74 | 96.69 | 95.90 | 97.20 | 50.78 |
| GPT Image-1 (OpenAI, 2025) | - | 59.41 | 48.04 | 40.40 | 53.92 | 97.50 | 98.85 | 97.69 | 63.07 |

resenting the rapid evolution of the field: (A) **Classical & Flow-based Diffusion Models:** This group establishes our fundamental baselines and tracks the evolution, ranging from foundational models such as SD1.5 (Rombach et al., 2022) and SD-XL (Podell et al., 2024) to recent high-performance open-sourced models such as SD3.5 Large (Esser et al., 2024) and FLUX.1-dev (Labs, 2024). (B) **Long-Prompt Optimized Models:** We evaluate models specifically proposed to enhance complex instruction following built upon SD1.5 or SD-XL backbones. This includes LLM4GEN (Liu et al., 2025), LLM Blueprint (Gani et al., 2024), ELLA (Hu et al., 2024), LongAlign (Liu et al., 2024), and ParaDiffusion (Wu et al., 2025c). (C) **Models with an LLM/MLLM Encoder:** This category includes advanced models like SANA (Xie et al., 2024) and Qwen-Image (Wu et al., 2025b), which replace traditional text encoders (e.g., CLIP) with powerful ones to achieve deeper semantic extraction. (D) **Autoregressive Models & Unified Models:** We examine recent architectural paradigm shifts, including the Janus family (Janus (Wu et al., 2025a), JanusFlow (Ma et al., 2025), Janus-Pro (Chen et al., 2025b)), Lumina-Image-2.0 (Qin et al., 2025a), and BAGEL (Deng et al., 2025). These models leverage autoregressive generation, Mixture-of-Transformers (MoT) designs, or unified architectures to better align text and visual patches. (E) **Proprietary SOTA Models:** To establish the absolute upper bound of current capabilities, we include leading proprietary models: Gemini 2.0 Flash (Kampf & Brichtova, 2025) and GPT

Image-1 (OpenAI, 2025). In Appendix Q, we also evaluate the recently released Nano Banana 2 (DeepMind, 2026) and discuss the performance differences and underlying reasons compared to the other SOTA models.

## 4.2. Performance Evaluations on DETAILMASTER

### 4.2.1. COMPARISONS ACROSS GENERAL-PURPOSE TEXT-TO-IMAGE MODELS

**Impact of token length constraints.** As shown in Table 2, models employing CLIP as the text encoder (SD1.5 and SD-XL) show significant limitations when handling long prompts due to token length constraints, consistently underperforming across all four tasks. In contrast, DeepFloyd IF, which employs T5 as its text encoder, benefits from an extended input capacity and visibly outperforms the former two models. Nevertheless, its performance remains unsatisfactory, constrained by both the quality of its short training data and inherent architectural limitations.

**Performance ceiling of SOTA models.** SD3.5 and FLUX, which combine T5 with enhanced training data and superior architecture, show improvements over previous models. They achieve over 90% accuracy on "Scene Attributes", highlighting strong scene control capability. However, their performance remains constrained on the other three tasks, suggesting that the absence of explicit training on long prompts limits their adaptability. Additionally, proprietary

models Gemini 2.0 Flash and GPT Image-1 outperform all aforementioned open-source models. Nevertheless, their accuracy for "Character Attributes", "Character Locations", and "Entity Relationships" plateaus around 50%, indicating that even SOTA models have considerable room for improvement in long prompt scenarios.

### 4.2.2. COMPARISONS ACROSS LONG-PROMPT OPTIMIZED TEXT-TO-IMAGE MODELS

**Long prompt training matters more than increasing token capacity.** While LLM4GEN uses an adapter to infuse T5 features, its 128-token limit during training fails to address long prompt challenges, leading to slight improvements. ELLA maintains the same token constraint but train with long and complex prompt, yielding more improvements than LLM4GEN. With a similar architecture and training strategy, ParaDiffusion extends the limit to 512 during training, enabling superior performance. Compared to DeepFloyd IF (with same token capacity but conventional training), ParaDiffusion shows better performance across almost all metrics. These results indicate that while expanded token capacity provides necessary infrastructure, long prompt training yields greater gains.

**Decomposition and iteration mitigates long-prompt challenges.** LongAlign decomposes long prompts for separate encoding and trains with long prompts, indirectly increasing the token capacity and achieving notable gains. LLM Blueprint identifies key characters and locations from long prompts, followed by iterative image refinement, yielding notable improvements in "Character Locations".

**Performance bottlenecks in backbone architectures.** Nevertheless, these long-prompt optimized methods are predominantly built upon SD1.5 and SD-XL. While these methods show improvements over their baseline counterparts, their overall performance remains unsatisfactory.

**Advanced LLM/MLLM Encoders:** Integrating a stronger encoder significantly improves semantic extraction. Models such as SANA and Qwen-Image show substantial leaps in Character Attributes compared to early diffusion models. However, our results indicate that better encoders alone are insufficient for complete prompt adherence. Even these advanced models experience performance drops on the hardest dimensions (i.e., Character Locations and Entity Relationships). The gap between semantic understanding and spatial/relational rendering highlights a persistent challenge in current architectures. More details are in Appendix O.

**Autoregressive Models & Unified Models:** This category reveals the benefits of architectural innovation. Models such as the Janus family, Lumina-Image-2.0, and BAGEL-7B-MoT show robust performance across the board. We attribute their success under long, complex prompts to three factors: (1) specific architectural choices, such as decoupled encoders or MoT (Mixture-of-Transformers) style designs that better align textual concepts with visual patches; (2) richer training data at scale (e.g., UniCap); and (3) larger overall model scale. More details are shown in Appendix P.

### 4.2.3. ANALYSIS ON KEY GENERATION TASKS

**Error source analysis.** Our analysis of failure cases (detailed in Appendix R) illuminates the model's primary weaknesses. The predominant error patterns are: 1) conflicts between spatial instructions and the model's real-world positional priors; 2) difficulty in rendering complex descriptions, including detailed character interactions and intricate apparel; 3) cascading failures, where initial errors in character attributes preclude the correct generation of entity relationships; 4) errors in scene attributes, typically confined to excessive structural complexity or highly specialized styles.

**Style degradation from optimization.** Interestingly, for the style attribute, the optimized models underperform. This may stem from our benchmark's bias towards the realistic styles: while SD1.5 naturally aligns with real-world styles, optimizations may introduce undesirable deviations.

**Stronger models mitigate attribute misalignment.** In Appendix S, we assess attribute accuracy only for present characters to isolate attribute misalignment from character omission. Results show longer prompts increase attribute leakage, with advanced models showing less degradation.

### 4.3. Correlation between Length and Adherence

To better understand the impact of prompt length on T2I generation, we analyze the relationship between prompt token length (tokenized using CLIP's tokenizer) and the accuracy of "Character Attributes", "Character Locations", and "Entity Relationships". We employ five distinct length intervals: below 250, above 400, and three 50-token intervals spanning 250-400. The results, as illustrated in Figure 3 and 7, show a consistent negative correlation between prompt length and generation accuracy across 10 evaluated models (excluding SD1.5 and SD-XL due to their limited context windows). This indicates that current T2I models indeed struggle to faithfully follow long, complex prompts.

Furthermore, our analysis in Appendix T reveals this is not merely an issue of character omission. Even for the present characters, the fidelity of attribute alignment also deteriorates with increasing prompt length, highlighting a failure in attribute binding under detail-intensive conditions.

### 4.4. From Empirical Results to Model Limitations

**Reliance on Advanced Encoders for Structural Parsing.** CLIP-based models fail in spatial and relational tasks (Character Locations & Entity Relationships), indicating

*Table 3.* Controlled ablation study on the prompt token limit and training prompt length.

| Model | Character Attributes | | | Character Locations | Scene Attributes | | | Entity Relationships |
| --- | --- | --- | --- | --- | --- | --- | --- | --- |
| | Object | Animal | Person | | Background | Light | Style | |
| (1) 77 Limit w/ Short-Prompt-Training | 24.38 | 28.17 | 16.42 | 9.39 | 29.31 | 74.72 | 46.64 | 8.43 |
| (2) 512 Limit w/ Short-Prompt-Training | 25.37 | 30.11 | 18.27 | 10.45 | 34.80 | 74.16 | 66.22 | 10.12 |
| (3) 77 Limit w/ Long-Prompt-Training | 27.44 | 31.54 | 18.32 | 10.93 | 33.70 | 76.62 | 53.72 | 10.66 |
| (4) 512 Limit w/ Long-Prompt-Training | **30.50** | **33.13** | **19.36** | **12.52** | **37.51** | **80.68** | **69.98** | **13.07** |

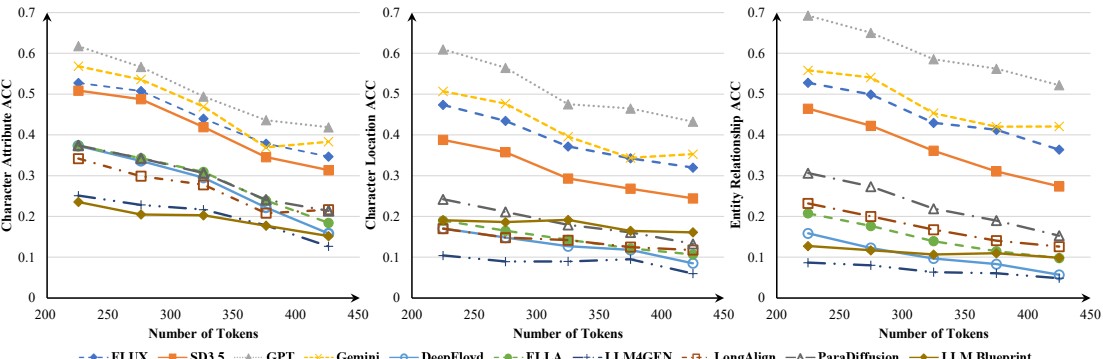

*Figure 3.* Negative correlation between generation accuracy and prompt token length.

that the employed encoders may struggle to parse structured prompts involving grammatical relations. However, as shown in Table 2 and Appendix O, models employing advanced LLM or MLLM encoders show substantial improvements in spatial layout. This performance gap verifies that capturing the syntactic dependencies (e.g., modifiers, clauses) in long prompts requires the superior linguistic processing capabilities inherent in advanced encoders.

**Attribute leakage.** The negative correlation between prompt length and adherence suggests that compositional complexity overwhelms the model's binding capacity. This is not solely due to character omission. As detailed in Appendix T, even for present characters, attribute fidelity decreases as prompts lengthen. This indicates that instead of being robustly bound, attributes are tend to leak, be omitted, or be incorrectly assigned to other objects in the image.

### 4.5. Ablation Study: Disentangling Token Constraint and Long Prompt Training

**Experimental setup.** To better compare the performance gains brought by various strategies, we conduct a 2x2 ablation study varying two factors: prompt token limit (77 and 512) and training prompt type (Short and Long). More details are in Appendix C and results are in Table 3.

**Long-prompt training matters more.** The model trained on long prompts with a 77-token limit consistently outperforms the one trained on short prompts with a 512-token limit, demonstrating that simply increasing context capacity is insufficient. Explicit training on detail-rich, long-form text is essential for interpreting complex compositions. In

Appendix U, we further provide evidence that these compositional gains persist even evaluated on short prompts.

**Synergistic effect.** The fourth configuration achieves the best performance, which means that the two factors are synergistic. An expanded prompt limit provides the capacity to process long prompts without truncation, while long-prompt training teaches the model to leverage that capacity to generate images with higher fidelity to the detailed instructions.

## 5. Evaluation of Data Quality and Diversity

### 5.1. Data Statistics

**DETAILMASTER** contains 4,116 prompts, covering 5,165 distinct nouns with an average prompt length of 284.89 tokens. The token length distribution reveals: 285 prompts at 100-200, 2,399 prompts at 200-300, 1,151 prompts at 300-400, and 281 prompts exceed 400. Its comprehensive annotations include: 1) "Character Attributes": 8,597 characters annotated with 37,165 features (22,728 for "object", 4,810 for "animal", and 9,627 for "person"); 2) "Character Locations": 6,910 character position annotations; 3) "Scene Attributes": 4,104 background, 4,114 lighting, and 4,112 style. 4) "Entity Relationships": 18,526 spatial/interactive relationship annotations. As shown in Table 1, our benchmark surpasses existing benchmarks through more comprehensive metrics and longer prompts.

### 5.2. Human Evaluation

To assess the data quality of **DETAILMASTER**, we randomly select 50 samples from each evaluation task (400 in

total) and employ two expert annotators to evaluate them, with scores being averaged. The evaluation is based on three criteria: 1) **Task Relevance:** alignment of the annotations with their designated tasks; 2) **Constraint Grounding Correctness:** alignment of the annotations with the source images; 3) **Prompt Consistency:** whether the annotations are reflected in the final polished prompts. The results show that 100% of samples meet task relevance requirements, 93.6% maintain visual fidelity with source images, and 97.5% show complete consistency with final polished prompts. These findings suggest that our benchmark is accurately derived from authentic image-caption pairs, and the close alignment between the final prompts and the annotations reduces evaluation errors resulting from missing attribute descriptions. These results collectively suggest the high quality and robustness of our benchmark. Detailed human evaluation results for specific sub-tasks are provided in Appendix G.

## 6. Further Analysis and Insights

**Benchmark validity and robustness.** We validate the high fidelity of **DETAILMASTER** through human evaluation, which shows near-perfect alignment between our prompts and annotations (Appendix G, V). This strictly resolves the information deficit in original captions, where constraints like character locations are missing in 93.26% of cases (Appendix I). Furthermore, the robustness of our LLM-based evaluation is validated by consistent model rankings and conclusions across a different evaluator and random seeds (Appendix J, L), ensuring that our findings are reproducible and not an artifact of the evaluator choice.

**A hierarchy of compositional failure.** Our fine-grained metrics reveal a clear difficulty hierarchy for T2I models. Precise spatial reasoning (*Character Locations*) is the most challenging task, which due to conflicts with strong real-world priors, followed by rendering complex *Person Attributes*. In contrast, general *Scene Attributes* such as lighting conditions and style elements are rendered with much higher fidelity. In Appendix R, our analysis reveals why this hierarchy exists. "Character Locations" is the most failure-prone because prompts often contradict pre-existing models' real-world priors. Models exhibit a stubborn tendency to generate objects in the center or in their canonical real-world placements. For "Person Attributes", failures primarily stem from complex clothing descriptions and intricate physical interactions, which easily trigger attribute leakage between the person and the interacted object.

**Attribute binding fails under high detail loads.** The negative correlation between prompt length and accuracy (see Figure 3) is not merely due to character omission. Our analysis shows that even for the present characters, attribute accuracy degrades as prompts grow longer (Appendix T).

**Further evaluations and analysis.** We evaluate models using powerful LLM/MLLM encoders and unified models, analyzing the impact of their improvement strategies across various dimensions, exploring promising directions for advancing T2I generation (Appendix O, P). Additionally, we extend our assessment to the recently released Nano Banana 2, providing insights into the evolving capabilities of frontier proprietary models and showing that mitigating strong visual priors and attribute leakage in long-prompt-driven T2I generation remains an challenge (Appendix Q).

**Fostering broader adoption.** To promote community adoption, we provide a resource-efficient *Mini-Benchmark* for rapid evaluation, along with a lightweight evaluator version compatible with limited GPU VRAM, which preserves the model ranking trends of main evaluation results (Appendix N, K). Furthermore, we validate the framework's compatibility with external metrics for assessing orthogonal dimensions such as aesthetics and safety (Appendix M).

## 7. Discussion and Conclusion

We introduce **DETAILMASTER**, a large-scale benchmark for evaluating compositional generation capabilities of T2I models on long, detail-rich prompts. Our evaluations with various models reveal model limitations and insights: even SOTA models exhibit deficiencies in complex compositional generation, with prompt adherence deteriorating as prompt length grows; the primary drivers of enhanced long-prompt adherence are not larger context windows but methods such as long-prompt training and iterative decomposition. We hope our work fosters progress toward more precise, controllable, and detail-oriented T2I models, unlocking applications hindered by the lack of a dedicated benchmark.

Our analysis reveals that failures in long-prompt scenarios are caused by text encoders that fail to capture syntactic dependencies within prompts, while diffusion models suffer from "attribute leakage." The rigorous investigation of these mechanisms remains an open question, pointing toward several promising avenues for future research: 1) **Structure-Aware Prompt Understanding:** Developing methods that explicitly model a prompt's syntactic or semantic graph structure, rather than treating it as a flat token sequence, to better preserve fine-grained meaning (Duan et al., 2024b). 2) **Object-Centric Diffusion Models:** Investigating architectures that learn disentangled, object-centric latent representations (perhaps via specialized attention or binding slots) to mitigate attribute leakage and enforce stricter object-attribute associations (Chen et al., 2026). 3) **Curriculum-Based and Data-Centric Training:** Using our data construction pipeline to create massive, high-quality datasets for long-prompt training, and exploring curriculum learning that progresses from simple to complex prompts to improve compositional generalization (Shen et al., 2026).

## Acknowledgement

This work was supported in part by the New Generation Artificial Intelligence-National Science and Technology Major Project (2025ZD0123003), and the National Natural Science Foundation of China Enterprise Innovation and Development Joint Fund (Artificial Intelligence Field) Key Support Projects (U25B2072).

## Impact Statement

This paper presents work whose goal is to advance the field of Machine Learning. There are many potential societal consequences of our work, none of which we feel must be specifically highlighted here.

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

# A. Overview of the Appendix

We provide more details and experiments of this work in the appendix and organize them as follows:

**1. Core Methodology and Implementation Details:**

- Appendix B, **Spatial Partitioning Scheme**: We introduce a custom Spatial Partitioning Scheme for categorizing bounding boxes into positional regions and empirically validate its superior performance over the conventional nine-grid partitioning method.

- Appendix C, **Configurations of Evaluated Models**: We detail the configurations of all evaluated models, including image resolutions and key inference hyperparameters.

- Appendix D, **Time Consumption and Resource Utilization**: We document the computational costs associated with dataset construction and model evaluation, including detailed time consumption and GPU resource utilization.

- Appendix E, **LLM Prompt Design for Our Pipeline**: We detail the specific prompt designs employed throughout our LLM-driven pipeline. The effectiveness of our data construction and model evaluation stages relies heavily on these carefully crafted prompts, which are engineered to elicit precise and reliable outputs from the LLM.

- Appendix F, **Evaluation Pipeline**: We provide a comprehensive technical specification of our evaluation pipeline, systematically detailing the methodologies employed to assess accuracy across four critical dimensions: "Character Attributes", "Character Locations", "Scene Attributes", and "Entity Relationships".

**2. Robustness of Our Data Construction and Evaluation Pipeline:**

- Appendix G, **Detailed Human Evaluation Results**: We present detailed human evaluation results, validating our benchmark's high reliability.

- Appendix H, **Mitigating the Impact of Inherent LLM Biases via Auxiliary Techniques**: We present an analysis of our pipeline against LLM-induced biases, empirically validating that our suite of auxiliary techniques ensures the objectivity and integrity of the data construction and evaluation pipeline.

- Appendix I, **Quantitative Analysis on Attribute Loss in Original Captions**: We conduct a systematic analysis to measure the rate at which original captions lack important fine-grained attributes.

- Appendix J, **Robustness of the Single-MLLM Evaluation Pipeline**: We validate the robustness of our MLLM-based evaluation by re-running evaluations with an alternative evaluator (InternVL3-9B). Results show that while absolute scores may vary, relative model rankings and overarching trends remain consistent. Additional evaluations using both Qwen-VL and InternVL evaluators on Qwen-Image suggest no self-enhancement bias from family-matched encoder-evaluator pairs.

- Appendix K, **On the Feasibility of Leveraging Small-Sized MLLMs for Evaluation**: To facilitate broader community adoption, we provide a lightweight evaluator (Qwen2.5-VL-3B-Instruct) for environments with limited GPU VRAM, which preserves the model ranking trends observed in our main evaluation.

- Appendix L, **Randomness Analysis for Evaluation with LLMs**: We present a randomness analysis for our LLM-based evaluation pipeline, empirically confirming the reproducibility and robustness of our evaluation pipeline through random seed experiments.

**3. Supplementary Resources and Materials:**

- Appendix M, **Compatibility of Our Framework with Other Evaluation Metrics**: We demonstrate the compatibility of our benchmark by empirically evaluating its outputs with established external metrics, confirming its utility as a specialized component within broader, multi-faceted evaluation pipelines.

- Appendix N, **Mini DETAILMASTER Benchmark**: We introduces a mini-benchmark (800 detail-rich long prompts) that maintains evaluation consistency with the full benchmark while significantly reducing resource requirements, enabling rapid model assessment.

**4. Further Analysis and Insights:**

- Appendix O, **A Superior LLM/MLLM Encoder Yields Enhancements in Generative Output**: We evaluate T2I models that utilize a superior LLM or MLLM as the encoder (SANA and Qwen-Image), validating that this approach yields enhancements in generative output. Furthermore, we also provide a discussion of their respective performance gains and shortcomings across various metrics.

- Appendix P, **Performance of Unified Models on DETAILMASTER**: We conduct an evaluation of several advanced unified models, including Janus, JanusFlow, Janus-Pro, Lumina-Image-2.0, and BAGEL, to compare the performance gains achieved by their respective improvement strategies.

- Appendix Q, **Performance of a More Advanced Proprietary Model**: We evaluate the recently released proprietary model, Nano Banana 2, benchmarking its performance against leading open-source and frontier closed-source models. While it shows competitive compositional capabilities and outperforms advanced open-source models, our analysis reveals that it remains susceptible to real-world visual priors and attribute leakage, highlighting that faithful long-prompt adherence remains an open challenge even for the latest generation of models.

- Appendix R, **Statistical Analysis of Metric Difficulty**: We present a systematic analysis of metric difficulty, quantitatively establishing a clear hierarchy from Character Locations (most difficult) to Scene Attributes (easiest). Furthermore, our qualitative investigation of failure cases reveals common error patterns that illuminate the underlying weaknesses of the models.

- Appendix S, **Stronger Models Mitigate Attribute Misalignment**: We evaluate the percentage of correctly rendered attributes among successfully detected characters. This analysis isolates specific attribute binding performance, revealing that advanced models maintain higher precision even when controlling for character presence.

- Appendix T, **Negative Correlation between Prompt Length and Attribute Alignment**: We present the correlation analysis between prompt length and attribute accuracy in generated characters, demonstrating a consistent negative relationship between prompt length and attribute alignment fidelity.

- Appendix U, **Performance Robustness of Our Ablation Models on Short Prompts:** We evaluate the fine-tuned models in our ablation study on a short-prompt benchmark and verify that training on complex data successfully enhances the model's overall compositional capability, even when applied to short prompts.

- Appendix V, **Details on the Validation Process for High-Quality Prompts:** We present the verification steps implemented to ensure the high quality of the final prompts and annotations, along with corresponding examples.

- Appendix W, **Case Studies**: We present representative case studies, the conclusions of which align with our comparative evaluations in the main text.

## B. Spatial Partitioning Scheme

In this section, we introduce our Spatial Partitioning Scheme, a method employed in the attribute extraction pipeline to categorize bounding boxes into approximate positional regions (e.g., "the upper part of the image"). This strategy is introduced because, in practical use, raw bounding box coordinates are typically excluded from prompts in favor of positional descriptions.

To maximize the extraction of character location features, we deviate from the conventional 3×3 grid partitioning, as we observe that characters often lie near region boundaries, making conventional region classification difficult. Instead, we introduce slight overlaps between adjacent regions by expanding their boundaries, ensuring better character coverage within each region. This adjustment enhances the assignment of positional labels, ultimately enriching the extracted "Character Locations" features and resulting in prompts with more character location information.

Notably, this partitioning scheme is applied exclusively during attribute extraction to augment the quantity of "Character Locations" data. However, during evaluation, we do not apply this scheme and directly assess positional accuracy by comparing character locations (bounding boxes) against prompt content using the MLLM to ensure precise measurement.

Our spatial partitioning scheme defines nine distinct regions within a normalized 1×1 coordinate space, where each region is represented as $[x_0, y_0, x_1, y_1]$ with $(x_0, y_0)$ denoting the top-left coordinates and $(x_1, y_1)$ denoting the bottom-right

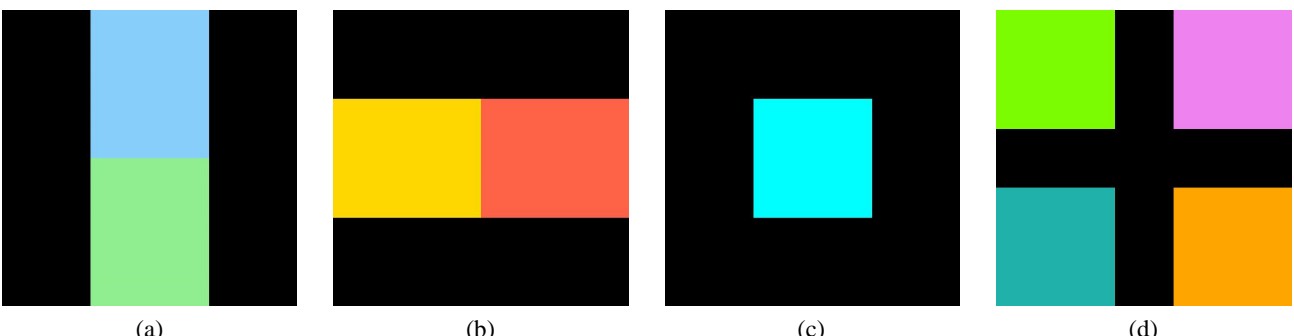

*Figure 4.* The visualization of the nine grids: (a) shows the upper part and the lower part; (b) shows the left part and the right part; (c) shows the middle part; while (d) shows all four corner regions: the upper left part, the lower left part, the upper right part, and the lower right part.

coordinates. Specifically, we define: the upper part as [0.3, 0, 0.7, 0.5], the lower part as [0.3, 0.5, 0.7, 1], the left part as [0, 0.3, 0.5, 0.7], the right part as [0.5, 0.3, 1, 0.7], and the middle part as [0.3, 0.3, 0.7, 0.7]. Additionally, we establish four corner regions: the upper left [0, 0, 0.4, 0.4], lower left [0, 0.6, 0.4, 1], upper right [0.6, 0, 1, 0.4], and lower right [0.6, 0.6, 1, 1]. As illustrated in Figure 4, this partitioning scheme provides systematic coverage of the image space while maintaining clear semantic correspondence with spatial relationships. With these nine regions, we first normalize the character's bounding box coordinates, then calculate the proportional area overlap between the normalized box and each of the nine spatial regions. The final position classification is determined by selecting the region that covers at least 75% of the character's bounding box area and shows the highest proportional overlap among all qualifying regions. For cases where no region meets the 75% coverage threshold (indicating boundary straddling), we assign a "null" location designation.

*Table 4.* Compared with the conventional 3×3 grid method, our partitioning scheme extracts richer valid character location labels and maintains matching accuracy with actual character locations.

|  | Proportion of "null" Character Location Labels | Matching Rate Between Labels and Actual Locations |
| --- | --- | --- |
| Conventional 3x3 Grid Method | 66.19% | 95.91% |
| Our Partitioning Scheme | 32.68% | 95.52% |

To validate the effectiveness of our partitioning scheme, we conduct a comparative analysis between the conventional 3×3 grid method and our proposed scheme by examining the proportion of ambiguous character positions (marked as "null") at region boundaries. The experimental results show: while the traditional method classify 7,821 out of 11,816 characters (66.19%) as "null" due to boundary ambiguity, our partitioning scheme reduce this number to only 3,862 (32.68%). This reduction provides empirical evidence that our scheme effectively prevents characters from being situated near region boundaries and extracts more valid positional descriptions.

Additionally, we employ the MLLM Qwen2.5-VL-7B-Instruct to verify the consistency between our position descriptions and actual character locations, following the same method as our evaluation pipeline: we highlight characters with red bounding boxes and provide both the box coordinates and the position descriptions to the MLLM for verification. Comparative analysis reveal that while the conventional 3×3 grid method achieved 95.91% accuracy, our proposed method attain 95.52% accuracy. This marginal difference shows that our scheme successfully maintains comparable localization accuracy while significantly increasing the number of detectable character positions, suggesting that our partitioning strategy achieves an optimal balance between precision and coverage.

## C. Configurations of Evaluated Models

### C.1. Inference Configurations

In this section, we detail the configurations for image generation across different models in our evaluation. Regarding the output resolution, we adopt 512×512 pixels for SD1.5 and its derivative models to align with their training settings, while setting 1024×1024 pixels for all other models. For inference steps (num_inference_steps), we follow each model's reference implementations: 100 steps for DeepFloyd IF and ParaDiffusion, 70 steps for ELLA, and 50 steps for remaining

models. The guidance scale parameters are similarly configured according to official recommendations: 11 for ELLA, 7 for DeepFloyd IF, 5 for SD-XL, 3.5 for both FLUX and SD3.5, and 7.5 for all other models.

Regarding the two proprietary models, Gemini 2.0 Flash and GPT Image-1, we access their official APIs, specifically "gemini-2.0-flash-exp-image-generation" and "gpt-image-1" respectively, setting the default hyperparameters for image generation.

For LLM Blueprint, we employ Qwen2.5-7B-Instruct as the LLM to perform attribute extraction and position extraction from long prompts. For Deepfloyd IF, "IF-I-XL-v1.0" serves as the first-stage model, "IF-II-L-v1.0" serves as the second-stage model, and "stable-diffusion-x4-upscaler" serves as the upscaler.

For each long-prompt optimized model, we employ the highest-performing open-source backbone available. It should be noted that while superior backbones have been described in their papers, their unavailability in open-source repositories precludes their inclusion in our evaluations.

### C.2. Experimental Setup for Ablation Study

On the experimental design for the ablation study described in Section 4.5, we modify the SD-XL model by integrating a T5 text encoder alongside the original CLIP encoders. To fuse the features from the T5 encoder, we adopt a Timestep-Aware Semantic Connector, similar to the mechanism used in ELLA. This structure allows the model to leverage the extended prompt limit and rich semantic representation of T5 without discarding the foundational capabilities learned from CLIP features. For efficiency, we keep the weights of the original SD-XL UNet, CLIP, and T5 frozen, only training the new semantic connector. As for the training data, we curate a training dataset of 2 million image-detailed caption pairs sampled from the DOCCI and Localized Narratives datasets.

We design four distinct experimental settings to systematically evaluate the impact of prompt limit and training prompt length: 1) **77 Token Limit w/ Short-Prompt Training.** The model is trained with a 77-token prompt limit using short prompts. 2) **512 Token Limit w/ Short-Prompt Training.** The model is trained with an expanded 512-token prompt limit, but still using short prompts. 3) **77 Token Limit w/ Long-Prompt Training.** The model is trained with a 77-token prompt limit using the detail-rich long prompts. 4) **512 Token Limit w/ Long-Prompt Training.** The model is trained with a 512-token prompt limit and our long prompts. Herein, the short prompts are generated by using Qwen2.5-VL-7B-Instruct to compress the original detailed captions to a length of 30 tokens or less, preserving the core semantic content.

## D. Time Consumption and Resource Utilization

In this section, we present the computational requirements of our data construction pipeline and evaluation pipeline.

For the first stage of our data construction pipeline (using Qwen2.5-VL-7B-Instruct and Qwen2.5-14B-Instruct), processing 164K detailed captions to produce 4,565 refined prompts on a single NVIDIA L20 GPU requires: 55 hours for "Main character identification", 24 hours for "Character localization", 77 hours for "Character attribute extraction", 36 hours for "Scene attribute and entity relationship detection", 55 hours for "Prompt refinement", and 59 hours for "Prompt filtering".

For the second stage of our data construction pipeline (i.e. "Prompt enhancement" with Qwen2.5-VL-72B-Instruct), processing 4,565 prompts derived from the first stage to produce 4,116 final polished prompts using 8 NVIDIA L20 GPUs requires: 3.3 hours for "Main character identification", 6 hours for "Character localization", 27.4 hours for "Character attribute extraction", 12.9 hours for "Scene attribute and entity relationship detection", 22.6 hours for "Prompt refinement", and 22.2 hours for "Prompt filtering".

The evaluation pipeline typically consumes 10 hours using a single NVIDIA L20 GPU, with duration positively correlated with the quality of generated images.

Regarding the GPU memory, both our data construction pipeline's first stage and the evaluation pipeline require 20GB-39GB of GPU memory. When executing the second stage of our data construction pipeline across 8 GPUs, each GPU shows memory usage ranging from 26GB to 38GB. The variation in memory requirement arises due to the different values of the "cache_max_entry_count" parameter, ranging from 0.01 to 0.8. Additionally, the previously reported time consumption is measured with the "cache_max_entry_count" parameter set to 0.8.

## E. LLM Prompt Design for Our Pipeline

In this section, we present the prompt designs for the LLMs (or MLLMs) used in our data construction phases and model evaluation phases.

In the "Main Character Identification" stage of our data construction process, we employ a few-shot approach to construct prompts that guide the model to respond in a JSON format. The prompt informs the model that it will be provided with *"an image and its corresponding description"*, and its task is to *"identify and count the main characters in the image"*. During this extraction phase, the MLLM also captures initial modifiers for each character, which significantly reduces the occurrence of duplicate entries in the Character List. Statistical analysis shows that in 97.49% of the samples, no repeated characters are present, as different characters can be clearly distinguished based on their respective modifiers. During prompt tuning, we experimented with zero-shot prompting but observed unsatisfactory performance. We also tried omitting the character count, which similarly degraded results.

For the "Character Localization" stage, we use the prompt *"Please only provide the bounding box coordinates of the region 'CHARACTER' describes"*, where CHARACTER refers to the identified main character from the previous stage. While Qwen2.5-VL-Instruct generally adheres to this instruction, occasional localization errors lead us to deploy an open-set detector YOLOE-11L to vote for the correct bounding box coordinates.

When extracting character attributes, we provide the model with the relevant image captions and, when necessary, highlight or crop the target regions of the characters while also supplying examples of the attributes to be extracted. During prompt tuning, we evaluated two alternative approaches: one eliminating image captions and another removing cropping/highlighting operations. Both configurations resulted in fewer extracted attributes. Additionally, we ensure the fidelity of the extracted character attributes. We iterate through each attribute associated with a character and employ the MLLM for verification: *"Please analyze the main character in this image. Is 'TEMP_CHARACTERISTIC' one of its features?"* Here, TEMP_CHARACTERISTIC denotes the character attribute being processed in the current iteration.

In the "Prompt Refinement" stage, we iteratively refine the image captions by prompting the model to supplement missing attributes: *"Some of the CATEGORY attribute information in the image description is missing. Your task is to supplement the missing CATEGORY attribute information into the image description."* Here, CATEGORY denotes different attribute types. For the "Filtering" stage, we prompt the model to determine which attributes are absent from the caption: *"Your task is to determine whether the given image description includes the description of this particular CATEGORY feature of the main character."*

For evaluating "Character Attribute", we prompt the MLLM to analyze each attribute individually, using cropped images of the characters: *"Please analyze the main character in this image, specifically the CHARACTER. Please determine whether ATTRIBUTE is one of its characteristics or is associated with it."* Here, CHARACTER and ATTRIBUTE are the specific character name and the attribute being evaluated. In the "Character Location" evaluation, we highlight the image and provide positional coordinates, prompting the MLLM to assess whether the object's location matches the reference. Regarding the feature evaluation for the entire image, the MLLM directly judges based on the image content and attribute categories.

## F. Evaluation Pipeline

We design a multi-stage evaluation pipeline to rigorously assess compositional T2I generation. The pipeline initiates by instructing T2I models with prompts from **DETAILMASTER** to produce corresponding image sets, followed by automated main character detection within the images. Subsequently, we perform sequential evaluations across four critical dimensions: 1) "Character Attributes", 2) "Character Locations", 3) "Scene Attributes" , and 4) "Entity Relationships". Regarding the evaluator, we employ Qwen2.5-VL-7B-Instruct as the MLLM throughout the evaluation process. To enhance reusability, our data construction process and evaluation pipeline are also implemented as data processing operators and configurable data recipes within Data-Juicer (Chen et al., 2024a; 2025a).

This section provides comprehensive technical specifications of our evaluation pipeline, detailing the methodologies and validation mechanisms implemented for each evaluation dimensions, as illustrated in Figure 5.

**Character presence identification.** Initially, we implement a dual-model verification system for main character detection, where a character is deemed successfully generated only if its presence is confirmed by both an MLLM and the YOLOE-11L open-set detector. For the confirmed characters, we also record their box coordinates to assist the subsequent attribute

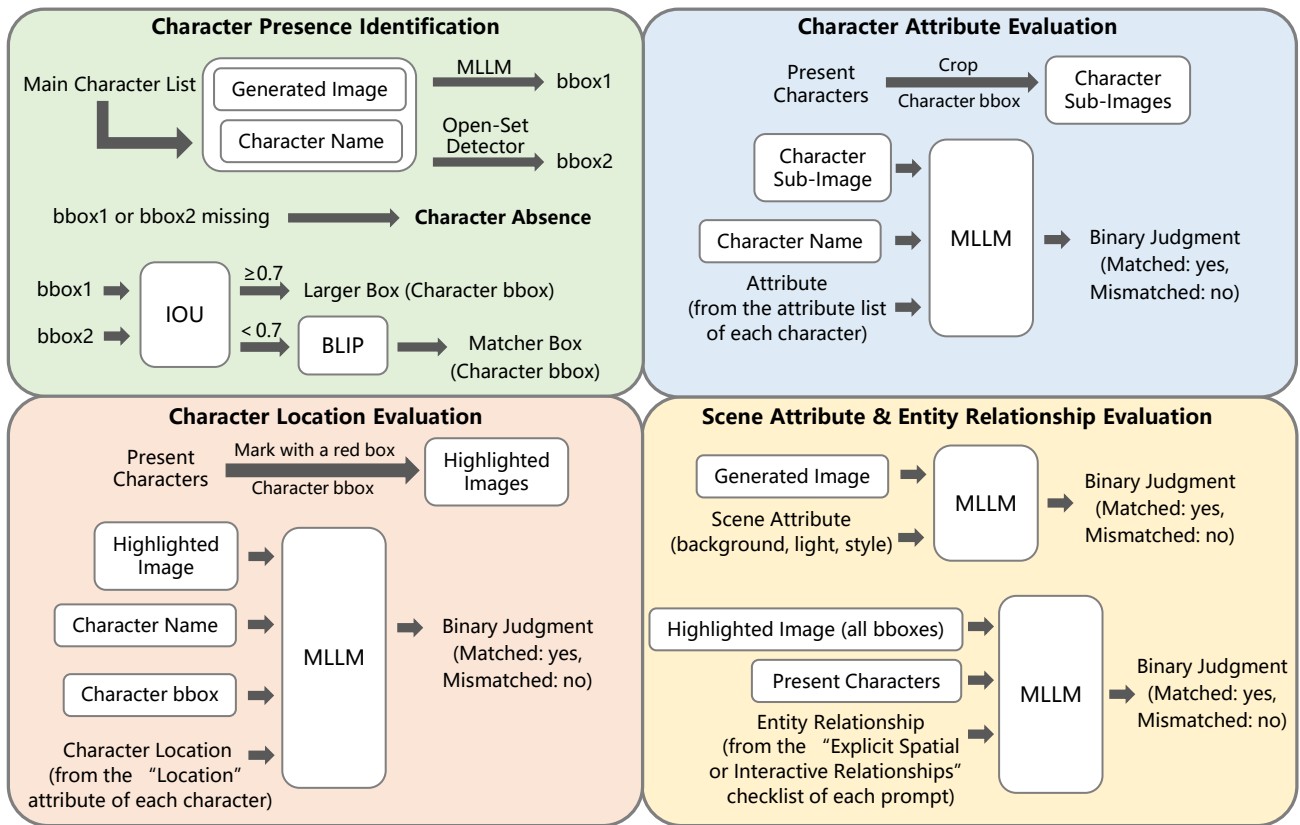

*Figure 5.* Overview diagram of the evaluation pipeline for the **DETAILMASTER** benchmark.

evaluation. Regarding how to determine the box coordinates, we calculate the IoU score between the two bounding boxes detected. If the score exceeds 0.7, we consider the boxes as overlapping and select the larger bounding box. Otherwise, we employ the vision-language model BLIP to compute image-text matching scores between both bounding box regions and the character name, selecting the box with a higher score. Identifying character presence can accelerate the whole evaluation process by eliminating redundant computations for failed generations. And it also avoids repetitive detection steps as the positional information can be reused.

**"Character Attribute" evaluation.** Our methodology for character attribute evaluation involves three main steps. First, we crop sub-images of present characters using their positional information. Second, each sub-image is processed by an MLLM, which renders a binary ("yes" or "no") judgment on the presence of each specified attribute. Third, we measure performance by calculating the accuracy of attribute generation. This metric is computed separately for objects, animals, and persons to allow for a detailed, category-level performance analysis.

**"Character Location" evaluation.** For character location evaluation, we sequentially process each identified character and mark its position in the generated image with a red bounding box. The bounding box information are then provided to the MLLM along with explicit instructions that the red box marks the target character. And the MLLM is prompted to determine whether the character location precisely aligns with the position description, responding with a yes/no judgment. The final character location accuracy is calculated as the proportion of correctly positioned characters to the total number of characters who have "Character Location" attribute.

**"Scene Attribute" evaluation & "Entity Relationship" evaluation.** For scene attribute evaluation, we ask the MLLM to determine whether the generated images satisfy each specified "Scene Attribute" with yes/no responses. The accuracy rates are calculated separately for background, lighting, and style attributes. Regarding entity relationship evaluation, for each generated image, we first mark all identified characters in the image with red bounding boxes. We then provide the MLLM with the marked image and the list of identified characters, and subsequently ask it to evaluate each attribute in the

"Spatial/Interactive Relationships" checklist, determining whether the specified entity relationship aligns with the image content.

**On the robustness of an MLLM-based evaluator.** While leveraging a single MLLM family (i.e., Qwen) for both data curation and evaluation could raise concerns about potential self-enhancement bias, we argue this risk is mitigated for two key reasons. First, our data construction is heavily grounded by auxiliary tools like open-set object detectors and guided by original human-annotated captions, breaking a purely end-to-end LLM pipeline. Second, as detailed in our robustness analysis (Appendix J), re-evaluating all models with a distinct MLLM (i.e., InternVL) preserves the relative model rankings and core conclusions. This indicates that **DETAILMASTER** measures a general compositional capability rather than an affinity for a specific model's idiosyncrasies.

## G. Detailed Human Evaluation Results

In this section, we provide the details of our human evaluation on **DETAILMASTER**, expanding the three metrics introduced in the main text (Task Relevance, Source Image Fidelity, and Prompt Consistency). Specifically, we systematically examine our benchmark based on the four key generation tasks.

*Table 5.* Human evaluation details for "Character Attributes".

| Category | Categorical Verification | Presence Confirmation | Prompt Inclusion | Attribute-Image Alignment | Attribute-Prompt Completeness |
|---|---|---|---|---|---|
| Object | 100% | 100% | 100% | 96.21% | 97.64% |
| Animal | 100% | 99% | 100% | 98.22% | 98.22% |
| Person | 100% | 98% | 99% | 91.58% | 96.17% |

For the "Character Attributes" (encompassing "object", "animal", and "person" categories), our annotators conduct a five-component examination for each designated main character: 1) **Categorical Verification** - determining whether the character correctly aligns with the annotated category; 2) **Presence Confirmation** - determining whether the character is in the source image; 3) **Prompt Inclusion** - determining whether the character is explicitly mentioned in the final polished prompt; 4) **Attribute-Image Alignment** - quantifying the proportion of annotated attributes that accurately align with the source image; and 5) **Attribute-Prompt Completeness** - quantifying the proportion of annotated attributes that are explicitly described in the final prompt. The results are shown in Table 5.

*Table 6.* Human evaluation details for "Character Locations".

| Presence Confirmation | Bounding Box Containment | Positional Accuracy | Prompt Inclusion |
|---|---|---|---|
| 100% | 99% | 100% | 99% |

For the "Character Locations", we employ a four-component spatial verification process: 1) **Presence Confirmation** - determining whether the character is in the source image; 2) **Bounding Box Containment** - determining whether the character falls in our annotated bounding box region (visually highlighted with a red rectangle); 3) **Positional Accuracy** - comparing the character's actual placement against our position annotations (e.g., "the upper left part of the image"); and 4) **Prompt Inclusion** - checking whether the spatial information in the final prompt matches our position annotations. The results are shown in Table 6.

*Table 7.* Human evaluation details for "Scene Attributes".

| Category | Categorical Verification | Scene_Attribute-Image Alignment | Prompt Inclusion |
|---|---|---|---|
| Background | 100% | 99% | 100% |
| Light | 100% | 91% | 99% |
| Style | 100% | 100% | 100% |

For the "Scene Attributes" (covering background, lighting, and style conditions), we involve three systematic checks: 1) **Categorical Verification** - validating that the annotated "scene_attribute_content" correctly describes the target visual information (i.e., background descriptions, lighting conditions, and style information); 2) **Scene_Attribute-Image Alignment** - determining whether the annotated "scene_attribute_content" aligns with the source image; and 3) **Prompt Inclusion** -

determining whether the annotated "scene_attribute_content" is unambiguously specified in the final prompt. The results are shown in Table 7.

*Table 8.* Human evaluation details for "Entity Relationships".

| Entity_Relationship-Image Alignment | Entity_Relationship-Prompt Completeness |
|---|---|
| 88.58% | 97.89% |

For the "Entity Relationship", we focus on two critical aspects: (a) **Entity_Relationship-Image Alignment** - quantifying the proportion of annotated entity relationships that correctly align with the source image; and (b) **Entity_Relationship-Prompt Completeness** - quantifying the proportion of annotated entity relationships that are explicitly described in the final prompt. The results are shown in Table 8.

As shown in the human evaluation results, the evaluation data of our benchmark achieves near-perfect performance across all metrics, with most scores approaching 100%, suggesting the high standard of our attribute extraction pipeline. However, a few metrics exhibit relatively lower scores, such as the Attribute-Image Alignment for the "Person" category in "Character Attributes" (91.58%), the Scene_Attribute-Image Alignment for the "Light" category in "Scene Attributes" (91%), and the Entity_Relationship-Image Alignment in "Entity Relationships" (88.58%). Notably, these metrics all pertain to the alignment between annotated features and the source images, and their failure to reach full scores suggests a few instances of hallucination during attribute extraction. Nevertheless, the corresponding "Prompt Inclusion" and "Prompt Completeness" metrics for these features consistently approach 100%, indicating that while rare discrepancies may occur between annotations and source image content, these features are still incorporated into the final prompts. This ensures that the evaluation accuracy remains unaffected, as the prompt-conditional generation faithfully reflects the annotated features rather than the features in the source images.

**DETAILMASTER** evaluates prompt-following fidelity for long, detail-rich instructions. Given a text prompt $p$ and a generated image $x \sim G(\cdot \mid p)$, our goal is to measure whether $x$ satisfies a set of atomic constraints $\mathcal{C}(p)$ derived from $p$. Importantly, while each prompt is constructed from a real image-caption pair as a semantic seed, the final prompt is not intended to be a faithful caption of the source image. Therefore, in **DETAILMASTER**, the prompt $p$ is the sole source of supervision at evaluation time: we do not score a model by matching the source image. The source image is used only during data curation to reduce annotation ambiguity.

## H. Mitigating the Impact of Inherent LLM Biases via Auxiliary Techniques

Our pipeline leverages advanced LLMs for data construction and model evaluation. A critical consideration, therefore, is the potential influence of inherent biases from these models. In this section, we address this concern and present supplementary experiments to demonstrate that our pipeline is robust against such biases. We argue that the extensive integration of auxiliary techniques at each stage effectively mitigates these potential influences, ensuring the quality and objectivity of our dataset and evaluation process.

### H.1. On Potential Bias in LLM-based Data Construction

Several studies have leveraged LLMs or MLLMs for data synthesis to facilitate the training and evaluation of various models, yielding improvements in both model performance and assessment outcomes (Zhou et al., 2024; Jiao et al., 2025). However, a primary concern is that sole reliance on LLMs for data generation could introduce inherent biases of the models. To counteract this, we integrate a suite of auxiliary techniques into our data construction pipeline, including open-set object detection, image cropping, image highlighting, and few-shot prompting. These techniques serve to ground the MLLM's outputs in objective image features rather than learned priors, enhancing the accuracy and diversity of the generated data. This multi-faceted approach ensures that our dataset is not exclusively dependent on the generative capabilities of any single LLM, thereby insulating it from model-specific biases.

Furthermore, our selection of MLLM is also a deliberate step in quality control. We benchmark three leading models, including Qwen2.5-VL-Instruct, InternVL2.5 (Chen et al., 2024b), and LLaVA-OneVision (Li et al., 2024b), and observe that Qwen2.5-VL-Instruct exhibits more rigorous logical reasoning and generates more precise outputs. Consequently, we select it as the primary MLLM for our data construction pipeline.

## H.2. On Potential Bias in LLM-based Evaluation

A second concern is a potential evaluation bias favoring models that use LLM-based text encoders, given that our data is constructed with LLMs. This concern is unfounded for two main reasons. First, the LLMs employed as text encoders in the evaluated models (e.g., Llama-2, T5-XL) are architecturally distinct and typically older than the LLM chosen for our data construction, Qwen2.5-VL-Instruct. (For the LLM Blueprint model specifically, while we employ Qwen2.5-7B-Instruct for object name extraction from prompts, this LLM functions only as a pre-processing component and is not the text encoder used during image generation.)

Second, as previously detailed, our extensive use of auxiliary techniques ensures the rich diversity and factual accuracy of our **DETAILMASTER** dataset. This effectively decouples the dataset from an over-reliance on the specific characteristics of Qwen2.5-VL-Instruct, minimizing the potential inherent biases that could favor a particular model architecture during evaluation.

## H.3. Experimental Validation

To empirically validate our claims, we conduct additional ablation studies analyzing the impact of our auxiliary techniques and the choice of LLMs on the data construction process.

**Impact of auxiliary techniques on data quality.** We first examine the effect of the deployed open-set object detector during the Character Localization stage. Our findings indicate that this step corrects 16.6% of bounding box coordinates proposed by the MLLM, enhancing localization accuracy.

Next, we investigate the consequences of omitting image cropping during the Character Attribute Extraction stage. Specifically, we employ Gemini-2.5-Flash (Comanici et al., 2025) to evaluate the accuracy of each generated annotation. While the accuracy remains relatively stable, we observe a 15.1% reduction in the number of extracted attributes. This demonstrates that our image cropping is crucial for enabling the MLLM to perform a more comprehensive and detailed analysis.

**Impact of MLLM choice on data generation.** As previously mentioned, Qwen2.5-VL-Instruct shows relatively strict inference behavior, which our new experiment further provides evidence. We replicate the Character Attribute Extraction pipeline using InternVL3-9B (Zhu et al., 2025) and evaluate the resulting annotations with Gemini-2.5-Flash. The results indicate a 3.4% decrease in accuracy. Moreover, due to format errors in InternVL3-9B's outputs (e.g., failure to adhere to list format), the quantity of generated attributes decreases by 24.5%.

In conclusion, this section has rigorously addressed the critical issue of LLM-induced bias within our pipeline. Through a series of targeted experiments, we have demonstrated conclusively that our auxiliary techniques is indispensable for achieving high data quality. These methods not only rectify potential inaccuracies but also enrich the detail of our generated data. Therefore, our pipeline is robust against LLM-specific biases, establishing the **DETAILMASTER** dataset as a reliable and impartial resource for the community. Additionally, the performance degradation observed when substituting our carefully selected MLLM further underscores the validity of our model selection.

# I. Quantitative Analysis on Attribute Loss in Original Captions

To investigate how the original captions from DOCCI and Localized Narratives aren't long enough and miss the important, desirable details, we conduct a systematic analysis to measure the ratio at which original captions lack important fine-grained attributes. Specifically, we calculate which attributes from our final attribute list are missing in the original captions and compute the corresponding proportions (termed as the Attribute Lost Rate).

*Table 9.* Statistics of missing attributes in original captions.

|  | Character Attributes | Character Locations | Scene Attributes | Entity Relationships |
|---|---|---|---|---|
| Attribute Lost Rate | 63.07% | 93.26% | 60.20% | 68.36% |

As shown in Table 9, we find that:

- The missing rate for "Character Locations" is as high as 93.26%, indicating that positional information is rarely included. This quantitatively proves that original captions are fundamentally insufficient for training or evaluating

precise control, validating the necessity of our prompt re-captioning pipeline. Our pipeline effectively addresses this by explicitly localizing characters and expanding the prompts with spatial context.

- The missing rate for "Entity Relationships" is 68.36%, suggesting that interactions between characters are frequently overlooked. Our method mitigates this by leveraging character identification and MLLM-based reasoning to extract and incorporate relational semantics.

- The missing rate for "Character Attributes" is 63.07%. Case studies reveal that these captions often include only a few salient attributes, lacking comprehensive and nuanced descriptions.

- The missing rate for "Scene Attributes" (e.g., background, lighting, style features) is 60.2%, showing that holistic scene characteristics are not consistently captured.

## J. Robustness of the Single-MLLM Evaluation Pipeline

It is a common practice in existing Text-to-Image benchmarks to employ a single MLLM for evaluation. For instance, T2I-CompBench utilizes only Minigpt-4 (Zhu et al., 2024) for feature verification, while DPG-Bench relies on a single mPLUG-large model (Li et al., 2022a) for characteristic assessment. There are two primary reasons why benchmark works typically deploy only a single MLLM: (1) Deploying multiple MLLMs consumes significant time and GPU computing resource; (2) Adapting prompts, auxiliary tools, or evaluation frameworks for different MLLMs would lead to overly cumbersome inference code.

Our selection of Qwen2.5-VL-7B-Instruct is based on comparisons. We evaluate it against InternVL2.5 (Chen et al., 2024b) and LLaVA-OneVision(Li et al., 2024b), and find that Qwen2.5-VL-7B-Instruct demonstrates stricter judgment criteria. It shows superior accuracy in identifying object features while exhibiting strong adherence to prescribed output formats. In contrast, both InternVL2.5 and LLaVA-OneVision exhibit a tendency to respond "yes" more readily during feature evaluation, which consequently leads to inflated scores.

**Robustness of model rankings across different MLLM evaluators.** We conduct a new set of experiments, re-running our entire evaluation pipeline using InternVL3-9B. The results are presented in Table 10.

*Table 10.* Evaluation results on the **DETAILMASTER** Benchmark (with InternVL3-9B).

| Model | Character Attributes | | | Character Locations | Scene Attributes | | | Entity Relationships |
|---|---|---|---|---|---|---|---|---|
| | Object | Animal | Person | | Background | Light | Style | |
| SD1.5 | 22.62 | 24.03 | 17.84 | 11.90 | 33.48 | 77.20 | 83.90 | 12.99 |
| SD-XL | 36.41 | 44.03 | 29.19 | 20.97 | 38.62 | 79.82 | 69.65 | 19.63 |
| DeepFloyd IF | 54.75 | 57.92 | 45.25 | 33.11 | 37.12 | 79.66 | 85.18 | 28.30 |
| SD3.5 Large | 67.57 | 78.99 | 55.38 | 59.26 | 95.66 | 95.11 | 96.28 | 67.18 |
| FLUX.1-dev | 71.42 | 79.42 | 55.61 | 67.13 | 98.85 | 98.78 | 96.01 | 74.41 |
| Gemini 2.0 Flash | 72.37 | 80.08 | 54.65 | 69.31 | 98.72 | 98.62 | 97.51 | 77.27 |
| GPT Image-1 | 73.44 | 83.81 | 59.02 | 69.95 | 99.51 | 99.11 | 95.75 | 82.51 |
| LLM4GEN | 24.25 | 30.76 | 21.13 | 14.66 | 35.82 | 77.86 | 55.06 | 16.23 |
| LLM Blueprint | 22.02 | 22.90 | 18.65 | 28.26 | 66.03 | 87.79 | 65.30 | 20.61 |
| ELLA | 41.92 | 49.76 | 34.93 | 27.99 | 57.82 | 84.44 | 42.66 | 26.74 |
| LongAlign | 40.54 | 47.52 | 26.53 | 25.98 | 86.26 | 93.00 | 84.58 | 31.61 |
| ParaDiffusion | 51.68 | 52.84 | 39.11 | 45.88 | 90.45 | 95.45 | 68.13 | 56.88 |

As shown in the table, with InternVL3-9B, all models show varying degrees of score improvement. This observation aligns with our previous experimental findings on InternVL2.5, suggesting that using InternVL3-9B for evaluation is not sufficiently rigorous, as it tends to favor "yes" responses when assessing ambiguous features.

However, despite the overall score improvement, the comparative rankings among the models and the general score trends remain unchanged. For general-purpose T2I models, more advanced models consistently achieve higher scores. Similarly, long-prompt optimized T2I models outperform the baseline models SD1.5 and SD-XL.

Regarding additional details, for instance:

- Top-tier proprietary models (GPT Image-1, Gemini 2.0 Flash) still outperform all other models, with more advanced models consistently achieving higher scores.

- Long-prompt-specific models (ParaDiffusion, LongAlign, ELLA) still show clear advantages over baselines like SD-XL on complex attributes.

- ParaDiffusion continues to excel in Character Locations and Entity Relationships compared to other long-prompt-specific models.

- We also compute the Kendall's Tau Correlation Coefficient between the evaluation results obtained with the two MLLMs. The correlation coefficients for each metric are as follows: 0.879, 0.909, 0.909, 0.909, 0.970, 1.000, 0.879, and 0.970. These consistently high values (all approaching 1.0) reflect a strong positive correlation between the two MLLM evaluators, confirming the consistent scoring trends across the evaluated models.

This cross-evaluator consistency provides strong evidence that our benchmark's conclusions are robust and not an artifact of the specific MLLM used.

**No self-enhancement bias from family-matched encoder and evaluator.** Additionally, the concern about potential self-enhancement bias, especially for models like Qwen-Image (Wu et al., 2025b) that share an architectural family with our primary MLLM evaluator (QwenVL), is a valid and crucial aspect of benchmark robustness. To directly address this, we conduct a new evaluation of Qwen-Image on our benchmark using both the original Qwen2.5-VL-7B-Instruct evaluator and the architecturally distinct InternVL3-9B evaluator. The results are presented in Table 11.

*Table 11.* Evaluation of Qwen-Image using Qwen2.5-VL-7B-Instruct (w/ QwenVL) and InternVL3-9B (w/ InternVL)

| Model | Character Attributes | | | Character Locations | Scene Attributes | | | Entity Relationships |
|---|---|---|---|---|---|---|---|---|
| | Object | Animal | Person | | Background | Light | Style | |
| Qwen-Image w/ QwenVL | 51.01 | 49.11 | 39.30 | 46.98 | 98.35 | 98.98 | 96.08 | 60.04 |
| Qwen-Image w/ InternVL | 69.64 | 84.50 | 62.17 | 77.54 | 99.56 | 99.66 | 96.45 | 86.55 |

As the table demonstrates, Qwen-Image's absolute scores are higher when assessed by the InternVL3-9B evaluator. This observation is consistent with our findings in Table 10, where we noted that InternVL3-9B exhibits a more permissive evaluation tendency, leading to a general score inflation across all tested models.

However, the more critical finding is the consistency of the model's relative performance. Qwen-Image's performance trend compared to other evaluated models remains unchanged. Specifically:

- **Preservation of relative ranking:** Qwen-Image's top-tier ranking relative to other diffusion models (such as GPT Image-1 and FLUX.1-dev) is preserved. It continues to demonstrate excellent performance, regardless of whether the evaluator is from the Qwen family or not (see Table 2 and Table 10).

- **Refuting self-enhancement bias:** If a significant self-enhancement bias were present, we would expect Qwen-Image's performance advantage to diminish or disappear when evaluated by a non-Qwen MLLM. Our results show the opposite: Qwen-Image's strong performance is consistently recognized by an independent evaluator. This indicates that its high scores are attributable to its genuine compositional generation capabilities rather than an artifact of the evaluation setup.

In summary, this new experiment provides direct evidence that our evaluation framework is robust and that Qwen-Image's strong performance on **DETAILMASTER** is not a result of self-enhancement bias.

## K. On the Feasibility of Leveraging Small-Sized MLLMs for Evaluation

The computational requirements of our MLLM-based evaluation, particularly the VRAM needed for Qwen2.5-VL-7B-Instruct, could present a barrier for some research teams. Introducing a more lightweight evaluator is a crucial step toward increasing the accessibility of our benchmark.

Fortunately, our evaluation pipeline's robustness is not solely dependent on the MLLM. We integrate a suite of auxiliary techniques (such as open-set object detectors and structured verification steps) that ground the evaluation in objective features and enhance its accuracy. This design significantly mitigates the risk of performance degradation when switching to a smaller evaluator model.

In this section, we conduct a new set of experiments, re-running our entire evaluation pipeline using a smaller evaluator, Qwen2.5-VL-3B-Instruct. The comprehensive results are presented in Table 12.

*Table 12.* Evaluation results on the **DETAILMASTER** Benchmark (with Qwen2.5-VL-3B-Instruct).

| Model | Character Attributes | | | Character Locations | Scene Attributes | | | Entity Relationships |
|---|---|---|---|---|---|---|---|---|
| | Object | Animal | Person | | Background | Light | Style | |
| SD1.5 | 20.23 | 28.76 | 13.39 | 11.62 | 32.24 | 80.99 | 80.57 | 10.00 |
| SD-XL | 24.08 | 29.16 | 17.91 | 15.51 | 36.48 | 83.18 | 69.09 | 15.85 |
| DeepFloyd IF | 29.57 | 39.39 | 27.22 | 19.70 | 34.20 | 83.36 | 85.71 | 17.73 |
| SD3.5 Large | 45.25 | 45.85 | 33.82 | 40.04 | 93.62 | 96.38 | 94.99 | 51.85 |
| FLUX.1-dev | 47.61 | 44.39 | 35.25 | 45.20 | 96.83 | 98.81 | 93.95 | 57.25 |
| Gemini 2.0 Flash | 48.88 | 46.75 | 34.33 | 49.98 | 97.53 | 98.34 | 96.32 | 59.82 |
| GPT Image-1 | 54.82 | 46.16 | 40.98 | 55.47 | 98.52 | 99.24 | 95.75 | 69.63 |
| LLM4GEN | 21.35 | 29.70 | 18.54 | 12.63 | 39.28 | 82.67 | 58.32 | 11.79 |
| LLM Blueprint | 20.34 | 26.16 | 14.73 | 23.82 | 65.61 | 89.95 | 66.62 | 20.13 |
| ELLA | 30.53 | 34.60 | 22.08 | 20.26 | 56.51 | 88.38 | 41.15 | 23.62 |
| LongAlign | 28.86 | 33.14 | 15.62 | 19.96 | 85.43 | 94.97 | 82.09 | 27.71 |
| ParaDiffusion | 33.65 | 34.43 | 24.40 | 25.73 | 89.38 | 96.35 | 65.87 | 36.12 |

As the results demonstrate, while the absolute scores differ slightly from the original evaluation, the comparative rankings among the models and the general score trends remain unchanged. For general-purpose models, more advanced models consistently achieve higher scores. Similarly, long-prompt optimized models continue to outperform their baselines (SD1.5 and SD-XL).

Regarding more specific details, we observe consistent trends that align with our original findings:

- A clear performance chasm persists among general-purpose models. A significant gap separates older models (SD1.5, SD-XL, DeepFloyd IF) from advanced ones (SD3.5 Large, FLUX.1-dev, etc.).

- ParaDiffusion maintains its superior performance in "Character Locations" and "Entity Relationships" compared to other long-prompt optimized models.

- The performance on Scene Attributes highlights nuanced differences. While top-tier general-purpose models achieve near-perfect scores, the long-prompt optimized models show varied success. For instance, LongAlign and ParaDiffusion demonstrate remarkable gains in Background and Light fidelity, whereas others like ELLA show more modest improvements, especially in the Style category.

- We further compute the Kendall's Tau Correlation Coefficient between the evaluation results produced by the two MLLMs. The correlation coefficients for each metric are: 0.9697, 0.9090, 1.0, 1.0, 0.9394, 0.9394, 0.9090, and 0.9394. These consistently high values indicate a strong positive correlation between the two evaluators, demonstrating that the evaluation results obtained with Qwen2.5-VL-3B-Instruct are highly consistent with those from the 7B model.

This high correlation in evaluation outcomes confirms that Qwen2.5-VL-3B-Instruct is a viable and resource-efficient alternative for our benchmark, preserving the integrity of the relative model comparisons.

## L. Randomness Analysis for Evaluation with LLMs

To assess the impact of randomness inherent in LLM-based evaluation, we conduct experiments by varying random seeds and setting the LLM temperature to 0.8, subsequently repeating evaluation across five representative models (SD1.5, LongAlign, LLM Blueprint, FLUX.1-dev, and GPT Image-1) to quantify stochastic variability. The results are shown in 13.

*Table 13.* Robustness evaluation of LLM-based assessment randomness.

| Model | Character Attributes | | | Character Locations | Scene Attributes | | | Spatial Attributes |
|---|---|---|---|---|---|---|---|---|
| | Object | Animal | Person | | Background | Light | Style | |
| FLUX.1-dev (1) | 51.47 | 45.83 | 34.91 | 41.57 | 95.77 | 97.05 | 94.81 | 47.49 |
| FLUX.1-dev (2) | 51.72 | 45.89 | 35.60 | 41.57 | 95.77 | 97.05 | 94.81 | 47.49 |
| FLUX.1-dev (3) | 51.53 | 45.85 | 34.97 | 41.57 | 95.77 | 97.05 | 94.81 | 47.45 |
| GPT Image-1 (1) | 59.47 | 48.12 | 40.43 | 53.93 | 97.50 | 98.85 | 97.69 | 63.03 |
| GPT Image-1 (2) | 59.47 | 48.12 | 40.43 | 53.93 | 97.50 | 98.85 | 97.69 | 63.03 |
| GPT Image-1 (3) | 59.47 | 48.12 | 40.43 | 53.93 | 97.50 | 98.85 | 97.69 | 63.03 |
| LLM Blueprint (1) | 21.41 | 27.01 | 13.91 | 18.44 | 50.89 | 77.57 | 64.24 | 11.60 |
| LLM Blueprint (2) | 21.42 | 26.97 | 13.85 | 18.46 | 50.89 | 77.57 | 64.24 | 11.65 |
| LLM Blueprint (3) | 21.32 | 26.97 | 13.67 | 18.46 | 50.89 | 77.57 | 64.24 | 11.62 |
| LongAlign (1) | 32.95 | 34.60 | 15.35 | 14.89 | 77.03 | 87.70 | 72.27 | 19.17 |
| LongAlign (2) | 32.95 | 34.60 | 15.35 | 14.89 | 77.03 | 87.70 | 72.27 | 19.09 |
| LongAlign (3) | 32.99 | 34.56 | 15.38 | 14.92 | 77.03 | 87.70 | 72.27 | 19.11 |
| SD1.5 (1) | 20.79 | 27.69 | 13.89 | 8.68 | 22.02 | 64.52 | 80.90 | 5.88 |
| SD1.5 (2) | 20.76 | 27.68 | 13.92 | 8.68 | 22.02 | 64.52 | 80.90 | 5.84 |
| SD1.5 (3) | 20.78 | 27.62 | 13.93 | 8.68 | 22.02 | 64.52 | 80.90 | 5.84 |

The presented results show that the randomness introduced by LLMs in our evaluation framework has minimal impact, with repeated assessments across all models showing negligible variations ($\Delta < 0.5\%$). This robust consistency strongly validates the stability and reliability of our evaluation pipeline, ensuring reproducible and dependable model comparisons regardless of inherent LLM randomization factors.

## M. Compatibility of Our Framework with Other Evaluation Metrics

Our work specifically focuses on the adherence of T2I models to detailed, long prompts. Consequently, metrics such as "visual quality" and "safety" fall outside the scope of our objectives. Nevertheless, our benchmark is fully compatible with other frameworks that assess "visual quality" and "safety" scores. The prompts and generated images from our dataset can be readily integrated into such frameworks to obtain these corresponding scores.

To demonstrate this compatibility, we evaluate our prompts and generated images using several established external metrics: (1) DiffSynth-Studio's ImageReward (for image quality), Aesthetic (for aesthetic scores), and HPSv2.1 and MPS (for human preference scores) (Xu et al., 2023; Duan et al., 2024a; Wu et al., 2023a; Zhang et al., 2024); and (2) Falcons-ai's nsfw_image_detection (for safety assessment) (Falcons.ai, 2023).

*Table 14.* Demonstration of compatibility with external evaluation frameworks.

| Model | ImageReward | Aesthetic | HPSv2.1 | MPS | nsfw_image_detection |
|---|---|---|---|---|---|
| SD-XL | 0.08 | 5.67 | 0.2831 | 9.69 | 0.0011 |
| FLUX.1-dev | 0.45 | 5.41 | 0.2921 | 9.89 | 0.0005 |
| ELLA | 0.23 | 5.52 | 0.2486 | 9.18 | 0.0021 |
| ParaDiffusion | 0.16 | 5.60 | 0.2733 | 8.48 | 0.0011 |

As shown in Table 14, our benchmark can be effectively combined with other evaluation frameworks to obtain other metric results. In our code repository, we will include hyperlinks to these external evaluation frameworks, providing users with the flexibility to choose additional metrics according to their specific needs.

## N. Mini DETAILMASTER Benchmark

To address computational resource constraints and facilitate rapid evaluation for researchers, we develop a mini version of our DETAILMASTER benchmark comprising 800 detail-rich long prompts. The construction methodology for the

mini benchmark includes a sampling approach across four key dimensions: 1) For "Character Attributes", we analyze the distribution of object, animal, and person entities in each sample, selecting candidates containing more than two entities of each category into respective attribute candidate pools (i.e., "Object Attributes", "Animal Attributes", "Person Attributes"); 2) For "Character Locations", samples featuring more than three localizable characters are included in the location candidate pool; 3) For "Scene Attributes", candidates are selected based on the presence of background descriptions, lighting conditions, or style specifications (i.e., "Background Attributes", "Lighting Attributes", "Style Attributes"); 4) For "Entity Relationships", samples are filtered by requiring a minimum of five spatial or interactive relationships. The final mini benchmark composition is achieved through balanced random sampling, extracting 100 prompts from each of these eight candidate pools to form a representative yet efficient evaluation set. We conduct a comprehensive re-evaluation of all 12 models discussed in the main text using our mini DETAILMASTER benchmark, with the comparative results presented in Table 15.

*Table 15.* Evaluation results on the **mini DETAILMASTER** Benchmark. All values in the table represent accuracy percentages.

| (A) General-Purpose Text-to-Image Model | | | | | | | | | |
|---|---|---|---|---|---|---|---|---|---|
| Model | Backbone | Character Attributes | | | Character Locations | Scene Attributes | | | Spatial Attributes |
| | | Object | Animal | Person | | Background | Light | Style | |
| SD1.5 | - | 14.76 | 21.81 | 9.85 | 7.11 | 20.11 | 65.62 | 85.94 | 6.03 |
| SD-XL | - | 18.48 | 21.92 | 11.58 | 10.87 | 26.36 | 69.01 | 70.05 | 8.22 |
| DeepFloyd | - | 22.40 | 23.85 | 20.77 | 14.72 | 24.52 | 68.23 | 86.98 | 12.60 |
| SD3.5 Large | - | 36.36 | 32.27 | 23.00 | 35.45 | 94.57 | 88.80 | 98.70 | 35.99 |
| FLUX.1-dev | - | 37.90 | 38.57 | 27.43 | 46.66 | 97.83 | 97.14 | 95.83 | 43.04 |
| Gemini 2.0 Flash | - | 47.21 | 36.82 | 27.14 | 46.15 | 97.67 | 96.88 | 98.91 | 46.35 |
| GPT Image-1 | - | 50.96 | 39.42 | 31.23 | 66.06 | 98.75 | 98.84 | 95.40 | 55.87 |
| (B) Long-Prompt Optimized Text-to-Image Model | | | | | | | | | |
| Model | Backbone | Character Attributes | | | Character Locations | Scene Attributes | | | Spatial Attributes |
| | | Object | Animal | Person | | Background | Light | Style | |
| LLM4GEN | SD1.5 | 16.47 | 23.84 | 15.48 | 7.53 | 23.64 | 67.71 | 51.56 | 6.30 |
| LLM Blueprint | SD1.5 | 15.57 | 23.51 | 14.29 | 17.93 | 54.46 | 76.67 | 53.03 | 10.40 |
| ELLA | SD1.5 | 28.09 | 24.96 | 16.64 | 14.46 | 51.36 | 68.75 | 36.20 | 12.13 |
| LongAlign | SD1.5 | 26.08 | 23.42 | 13.91 | 15.97 | 85.60 | 81.25 | 63.80 | 15.45 |
| ParaDiffusion | SDXL | 28.97 | 28.40 | 17.51 | 18.56 | 89.40 | 88.80 | 60.36 | 18.66 |

The comparative results in Table 15 show that the performance comparison and trends across models remain consistent between our mini DETAILMASTER benchmark and the full DETAILMASTER benchmark. As the mini benchmark contains more challenging samples selected for their greater detail complexity, the scores on the mini benchmark are generally lower than those on the full version. Crucially, the preserved consistency combined with improved evaluation efficiency enables researchers to more conveniently assess model performance in handling long prompts. Therefore, the mini benchmark serves as an efficient alternative for resource-constrained scenarios or time-sensitive evaluations, while maintaining comparable validity to the full benchmark.

## O. A Superior LLM/MLLM Encoder Yields Enhancements in Generative Output

The recent trend of using powerful LLM/MLLM as text encoders represents a significant advancement in T2I generation, particularly for enhancing text understanding. It is meaningful to evaluate these advanced models with our benchmark to gauge their progress in handling long, detail-intensive prompts.

In this section, we conduct a new set of experiments on two representative models that leverage LLM-based or MLLM-based encoders: SANA (Xie et al., 2024), which employs a Gemma-2 as its encoder, and Qwen-Image (Wu et al., 2025b), which utilizes Qwen2.5-VL. The evaluation results on our benchmark are shown in Table 16.

To provide a more comprehensive analysis, we compare these new results against the models evaluated in our main text:

- **Impact of advanced encoders:** Both SANA and Qwen-Image substantially outperform earlier open-source models that rely on CLIP or T5 encoders. For example, SANA's performance on "Entity Relationships" is more than double

*Table 16.* Evaluation results of SANA and Qwen-Image on the **DETAILMASTER** benchmark

| Model | Character Attributes | | | Character Locations | Scene Attributes | | | Entity Relationships |
|---|---|---|---|---|---|---|---|---|
| | Object | Animal | Person | | Background | Light | Style | |
| SANA | 40.79 | 38.88 | 24.74 | 22.91 | 89.80 | 94.51 | 73.34 | 29.56 |
| Qwen-Image | 51.01 | 49.11 | 39.30 | 46.98 | 98.35 | 98.98 | 96.08 | 60.04 |

that of DeepFloyd IF. This indicates that a stronger text encoder enhances semantic extraction capacity and plays a crucial role in parsing the complex compositional and relational details present in long prompts.

- **Analysis of SANA:** SANA shows a solid improvement over older architectures, particularly in scene-level attributes. However, its performance on fine-grained compositional tasks like "Character Attributes", "Character Locations," and "Entity Relationships" still lags behind top-tier models. This suggests that while a better encoder provides a stronger semantic foundation, other architectural components of the diffusion model (e.g., SANA's focus on efficiency with linear attention) and the training data composition remain critical factors in achieving higher level of detail adherence.

- **Analysis of Qwen-Image:** Qwen-Image's performance is particularly impressive, achieving results that are highly competitive with, and in some areas even surpass, the leading proprietary model, GPT Image-1. The performance analysis proceeds as follows: 1) **Superior Compositionality.** Qwen-Image achieves an "Entity Relationships" score of 60.04%, which is very close to GPT Image-1's 63.07% and significantly higher than FLUX.1-dev's 47.49%. This highlights its exceptional ability to understand and render complex spatial and interactive relationships between multiple characters. 2) **Strong Attribute Binding.** Its scores on "Character Attributes" (e.g., 49.11% for Animal, 39.30% for Person) are on par with or exceed those of previous SOTA models, indicating robust attribute binding even under high detail loads. 3) **SOTA-Level Performance.** Overall, Qwen-Image sets a new standard for open-source models on our benchmark and narrows the gap with the best-performing closed-source systems. Its strong performance across all four dimensions indicates that the combination of a powerful MLLM encoder (Qwen2.5-VL) and a robust diffusion backbone is a highly effective path toward mastering long-prompt generation.

These new experiments also underscore the unique contribution of our benchmark. Simpler benchmarks focusing on short prompts might not reveal such a clear performance gap between models with different encoders. However, **DETAILMASTER**, with its average prompt length of over 284 tokens and its fine-grained evaluation across four critical dimensions, provides the necessary complexity and granularity to quantify the benefits of advanced encoders and identify persistent challenges. It highlights that even with superior text understanding, the accuracy on the most difficult compositional dimensions (Character Locations and Entity Relationships) remains far from perfect. In addition, handling intricate, long-form instructions remains an unsolved problem.

## P. Performance of Unified Models on DETAILMASTER

To provide a clearer picture of how recent unified models perform on our benchmark, we conduct an evaluation on several advanced unified models, including Janus (Wu et al., 2025a), JanusFlow (Ma et al., 2025), Janus-Pro (Chen et al., 2025b), Lumina-Image-2.0 (Qin et al., 2025a), and BAGEL(Deng et al., 2025). The results are presented in Table 17:

*Table 17.* Evaluation results of unified models on the **DETAILMASTER** benchmark

| Model | Character Attributes | | | Character Locations | Scene Attributes | | | Entity Relationships |
|---|---|---|---|---|---|---|---|---|
| | Object | Animal | Person | | Background | Light | Style | |
| Janus-1.3B | 35.86 | 32.15 | 22.31 | 21.4 | 87.59 | 95.67 | 83.53 | 25.48 |
| JanusFlow-1.3B | 31.93 | 37.54 | 16.5 | 17.3 | 81.24 | 94.8 | 92.99 | 16.77 |
| Janus-Pro-1B | 40.55 | 41.44 | 26.1 | 26.66 | 92.39 | 96.42 | 88.78 | 32.3 |
| Lumina-Image-2.0 (2B) | 43.74 | 43.79 | 29.2 | 34.28 | 86.86 | 93.47 | 93.81 | 39.68 |
| BAGEL-7B-MoT | 47.04 | 46.35 | 33.8 | 39.59 | 97.41 | 97.99 | 94.68 | 49.43 |

Our analysis of these results reveals several insights into what drives performance on detail-rich, long-prompt generation:

- **Janus-1.3B:** The original Janus model establishes a solid baseline. Its performance validates its core design of decoupling the understanding and generation encoders, which helps mitigate task conflict. Additionally, its unified architecture inherently enables it to comprehend longer prompts. However, its capabilities are constrained by its 1.3B scale and initial training data, leading to moderate scores in complex dimensions like Character Locations and Entity Relationships.

- **JanusFlow-1.3B:** While JanusFlow achieves a high Style score (suggesting high-quality image aesthetics), it underperforms the original Janus in most attribute-binding and spatial tasks. This result suggests that while the flow mechanism can improve overall image quality, it may compromise the model's adherence to the fine-grained compositional details prevalent in DetailMaster's prompts.

- **Janus-Pro-1B:** Janus-Pro shows a significant improvement across all metrics compared to its predecessors. As discussed in its paper, the "Pro" version benefits from an optimized training strategy and substantially expanded training data. This directly enhances its ability to interpret and render the complex compositional requirements of our benchmark.

- **Lumina-Image-2.0:** Lumina-Image 2.0 continues this upward trend. Its strong performance, especially the notable jump in Character Locations and Entity Relationships, can be attributed to its two key innovations mentioned in their work: the Unified Next-DiT architecture and the Unified Captioner (UniCap). UniCap is specifically designed to produce high-quality, multi-granularity textual descriptions for T2I tasks. Training on this highly descriptive data aligns perfectly with the demands of our DetailMaster benchmark, enabling the model to better ground visual elements to detailed textual specifications.

- **BAGEL-7B-MoT:** BAGEL achieves the highest performance across all dimensions. Its success stems from a combination of factors discussed in its paper: (1) Model Scale: At 7B parameters, it has a significantly larger capacity for knowledge and reasoning. (2) MoT Architecture: It employs a Mixture-of-Transformers (MoT) architecture with distinct experts for understanding and generation. This design minimizes task interference more effectively than a simple encoder decoupling, allowing each expert to specialize. (3) Large-scale Interleaved Data: Its training on trillions of tokens of interleaved multi-modal data equips it with superior compositional reasoning.

In summary, these results demonstrate that performance on DetailMaster is strongly correlated with architectural choices that mitigate task conflict (e.g., decoupled encoders, MoT), the richness of the training data (e.g., UniCap), and overall model scale.

## Q. Performance of a More Advanced Proprietary Model

In addition to Gemini 2.0 Flash and GPT Image-1 evaluated in the main text, we subsequently tested a newer T2I proprietary model, Nano Banana 2 (DeepMind, 2026) (released in February 2026). Furthermore, during image generation, we did not constrain the model to a strict 1024×1024 resolution, allowing it to natively determine the optimal image resolution.

For context, we present its performance alongside the leading open-source and proprietary frontier models currently in our benchmark. The results are presented in Table 18.

*Table 18.* Evaluation results of Nano Banana 2 on the **DETAILMASTER** benchmark

| Model | Character Attributes | | | Character Locations | Scene Attributes | | | Entity Relationships |
|---|---|---|---|---|---|---|---|---|
| | Object | Animal | Person | | Background | Light | Style | |
| SD3.5 Large | 48.56 | 46.20 | 32.95 | 33.62 | 89.61 | 90.33 | 95.69 | 40.03 |
| FLUX.1-dev | 51.47 | 45.83 | 34.91 | 41.57 | 95.77 | 97.05 | 94.81 | 47.49 |
| Gemini 2.0 Flash | 55.44 | 47.84 | 34.23 | 44.74 | 96.69 | 95.90 | 97.20 | 50.78 |
| GPT Image-1 | 59.41 | 48.04 | 40.40 | 53.92 | 97.50 | 98.85 | 97.69 | 63.07 |
| Nano Banana 2 | 57.52 | 48.56 | 35.94 | 46.46 | 98.81 | 98.76 | 98.49 | 62.40 |

We can draw several key insights regarding Nano Banana 2's capabilities in handling long and detail-rich prompts:

- **Superiority Over Advanced Open-Source Models:** Nano Banana 2 outperforms the advanced open-source models, such as SD3.5 Large and FLUX.1-dev, particularly in the structural bottleneck areas of "Character Locations" (46.46%) and "Entity Relationships" (62.40%). This confirms its advanced semantic extraction capacity for complex spatial and relational clauses.

- **Competitive with Proprietary Frontier Models:** Nano Banana 2 operates at the same tier as the advanced closed-source models. When compared to Gemini 2.0 Flash, Nano Banana 2 shows slight improvements across almost all fine-grained compositional dimensions, including Object (+2.08), Person (+1.71), and Entity Relationships (+11.62).

- **Performance Gaps and Future Directions:** Despite Nano Banana 2 being one of the latest proprietary T2I models (released after GPT Image-1), its performance in long-prompt scenarios still lags behind GPT Image-1. We compare the generated images from Nano Banana 2 and GPT Image-1, revealing that Nano Banana 2 is more susceptible to the influence of prior knowledge (or aesthetic considerations). It tends to generate objects according to real-world positional distributions, employ tilted angles characteristic of authentic photography styles, or zoom out the camera, thereby diminishing the size of the main characters. Consequently, objects generated in this manner are more prone to positional errors. Furthermore, when prompted for multiple identical objects, Nano Banana 2 tends to over-generate the objects and distribute the features across them, whereas GPT Image-1 correctly binds all features to each instance. This behavior exemplifies the "attribute leakage" mechanism we discussed.

Therefore, this suggests that recent research may not have sufficiently prioritized long-prompt following optimization, and the improved generation quality of new models might be confined to shorter prompts. Consequently, handling intricate, long-form instructions and mitigating "attribute leakage" remains an unsolved problem, leaving room for improvement even for the most capable models in the community.

## R. Statistical Analysis of Metric Difficulty

In this section, we present an experimental analysis of the relative difficulty of our four metrics. Furthermore, we conduct a failure case analysis to investigate instances where the evaluated models perform inaccurately.

Specifically, for each sample, we calculate the proportion of correctly generated attributes for each evaluation metric. If a model achieves a correct proportion below 50% for a specific metric, we consider that metric hard for the model on that sample (e.g., if a sample's prompt contains eight "animal attributes" and the model generates fewer than four correct "animal attributes", we consider the "Animal" generation task hard for the model on that sample).

The evaluation metrics we examine include: "Object", "Animal", "Person", "Character Locations", "Scene Attributes", and "Entity Relationships". Notably, the "Character Attributes" metric in our main evaluation is split into "Object", "Animal", and "Person" for individual analysis. Meanwhile, the sub-metrics of "Scene Attributes" (i.e., "Background", "Light", and "Style") are aggregated.

The results present the proportion of hard samples for each metric. We also calculate the average value across all evaluated models, as presented in the "Average" row. The final outcomes are shown in Table 19.

As shown in the table, the most challenging metric is "Character Locations". We check cases where all models made errors on this metric and identify two main reasons for the failures: (1) The object positions described in the prompt may contradict conventional real-world positions in photos. For instance, when the prompt states that traffic lights are in the "lower part" of the image while they typically appear in the "upper part" in real photos, this discrepancy leads to positional misalignment. (2) The models exhibit a tendency to generate objects in the middle of the image, even when the prompt specifies left or right placement.

The second most challenging metric is "Person" under the "Character Attributes" metric. From the error cases, we observe two failure patterns: (1) descriptions of the person often involve interactions with other objects (e.g., "holding a cat"); and (2) the clothing descriptions for the person are complex.

Regarding "Entity Relationships", the error cases are often correlated with the "Character Attributes" issues. When an object's attributes contain excessive errors, the evaluator will not recognize the object's existence, leading to negative judgments. Additionally, errors occur when the object's position deviates from conventional real-world placements.

Regarding "Scene Attributes", this metric is relatively the simplest. Models typically generate accurate results as long as the

*Table 19.* Proportion of hard samples for each metric.

| Model | Character Attributes | | | Character Locations | Scene Attributes | Entity Relationships |
|---|---|---|---|---|---|---|
| | Object | Animal | Person | | | |
| SD1.5 | 82.6% | 75.7% | 88.2% | 95.8% | 38.6% | 98.1% |
| SD-XL | 79.0% | 74.0% | 85.2% | 94.2% | 40.8% | 95.7% |
| DeepFloyd IF | 72.1% | 64.8% | 77.1% | 92.3% | 33.3% | 94.6% |
| SD3.5 Large | 54.4% | 56.0% | 69.6% | 76.5% | 2.7% | 62.9% |
| FLUX.1-dev | 51.7% | 56.5% | 67.9% | 68.8% | 0.8% | 52.2% |
| Gemini 2.0 Flash | 47.1% | 56.2% | 68.9% | 65.8% | 0.6% | 47.4% |
| GPT Image-1 | 42.7% | 55.6% | 62.9% | 55.2% | 0.2% | 32.4% |
| LLM4GEN | 81.7% | 73.8% | 84.9% | 95.3% | 52.8% | 97.3% |
| LLM Blueprint | 82.0% | 74.5% | 87.8% | 88.3% | 29.4% | 94.5% |
| ELLA | 69.0% | 68.5% | 79.9% | 90.9% | 43.6% | 90.4% |
| LongAlign | 71.2% | 68.1% | 86.4% | 91.9% | 11.6% | 87.0% |
| ParaDiffusion | 68.3% | 68.6% | 80.1% | 88.1% | 9.8% | 80.0% |
| **Average** | 66.8% | 66.0% | 78.2% | 83.6% | 22.0% | 77.7% |

supported prompt length is sufficient. For erroneous cases, aside from prompt truncation issues, failures primarily occur when: (1) the background contains excessive structural details, or (2) the style descriptions are specialized or uncommon.

## S. Stronger Models Mitigate Attribute Misalignment

In this section, we assess attribute accuracy exclusively for successfully generated characters to isolate model performance on attribute misalignment from character omission. Specifically, we use the Character Attributes metric, encompassing the Object, Animal, and Person categories. For images where characters are successfully generated, we quantify the percentage of attributes specified in the prompt that are accurately rendered. The results are presented in Table 20.

*Table 20.* Attribute accuracy of detected generated characters across models.

| Character Attributes | SD-XL | DeepFloyd IF | SD3.5 Large | FLUX.1-dev | Gemini 2.0 Flash | GPT Image-1 |
|---|---|---|---|---|---|---|
| Object | 79.17 | 81.19 | 86.55 | 89.01 | 90.06 | 91.77 |
| Animal | 84.27 | 82.73 | 89.39 | 90.37 | 90.37 | 92.31 |
| Person | 80.65 | 83.35 | 89.10 | 91.06 | 92.46 | 94.10 |
| Character Attributes | SD1.5 | LLM4GEN | LLM Blueprint | ELLA | LongAlign | ParaDiffusion |
| Object | 74.66 | 75.62 | 72.01 | 82.06 | 82.58 | 83.22 |
| Animal | 80.30 | 81.46 | 79.91 | 84.55 | 84.87 | 84.29 |
| Person | 75.93 | 77.90 | 71.98 | 84.55 | 82.48 | 83.40 |

As shown in Table 20, the performance disparity among models aligns with the main results in Table 2. Advanced model architectures achieve superior accuracy, while long-prompt optimization techniques also yield improvements over their baselines, demonstrating enhanced prompt comprehension and more precise attribute-character alignment.

## T. Negative Correlation between Prompt Length and Attribute Alignment

In this section, we analyze the relationship between prompt length and attribute accuracy for the detected generated characters (categorized into objects, animals, and persons) to examine how the model's attribute adherence vary with increasing prompt length. As illustrated in Figure 6, the left, middle, and right subgraph respectively show the attribute accuracy trends for "object", "animal", and "person" characters.

The trends show a clear inverse correlation between prompt length and attribute accuracy, indicating that longer prompts lead to progressively greater deviations from the intended character descriptions, manifested as attribute misalignment and omission. Notably, the performance comparison reveals that models specifically trained with long prompts (e.g., ELLA and ParaDiffusion) exhibit more gradual accuracy degradation compared to models using conventional training like Deepfloyd,

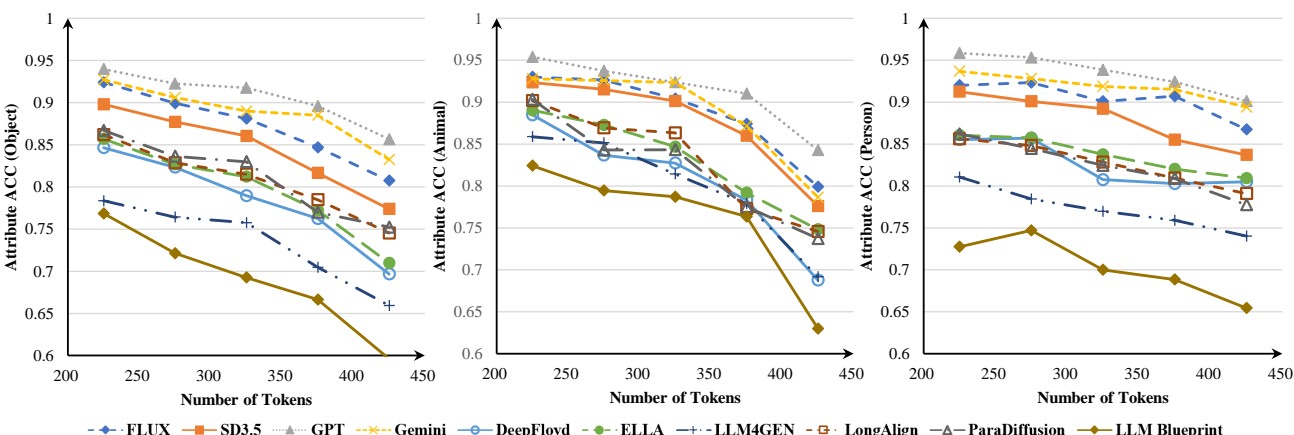

*Figure 6.* Negative correlation between prompt length and attribute alignment.

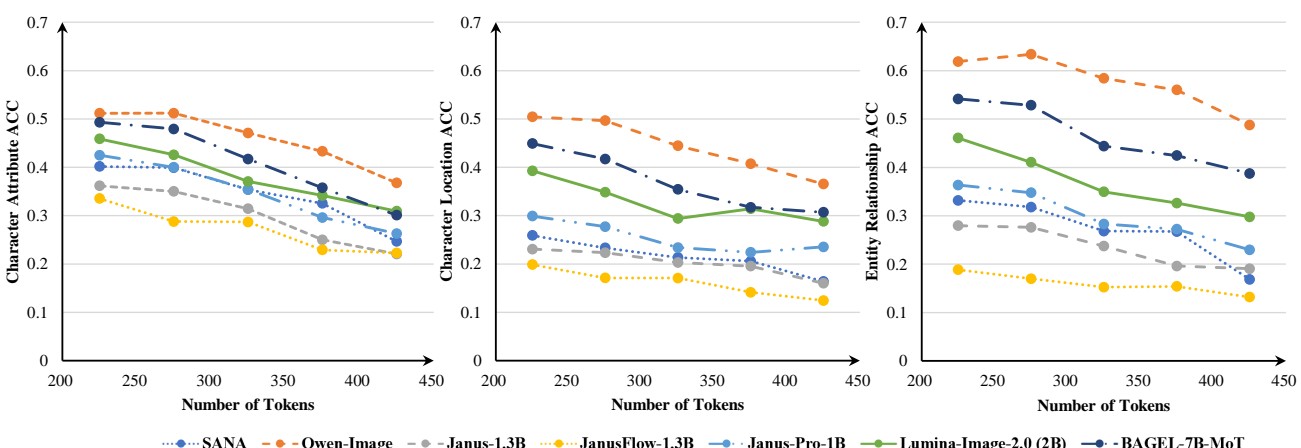

*Figure 7.* Negative correlation between generation accuracy and prompt length (Additional).

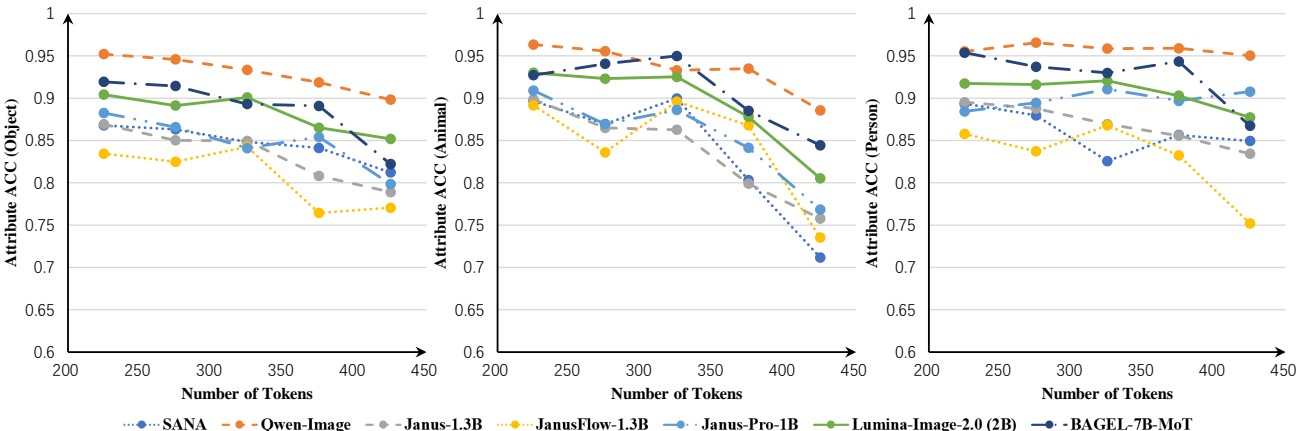

*Figure 8.* Negative correlation between prompt length and attribute alignment (Additional).

as evidenced by their shallower accuracy decline curves. These findings collectively suggest that current Text-to-Image generation still face substantial challenges in maintaining attribute consistency under detail-intensive conditions, highlighting the critical need for further research into long-prompt optimization techniques and training methodologies to advance this crucial capability.

Furthermore, we extend our analysis to include seven additional models presented in our appendix, examining the relationship between prompt length and both generation accuracy and attribute alignment. This new set includes MLLM-based encoder models (SANA, Qwen-Image) and unified models (Janus-1.3B, JanusFlow-1.3B, Janus-Pro-1B, Lumina-Image-2.0 (2B), BAGEL-7B-MoT). The results are presented in Figure 7 and Figure 8.

As illustrated in Figure 7, despite leveraging architectures better suited for long prompts, these models still exhibit a clear performance degradation as prompt length increases. However, Figure 8 reveals a crucial distinction in attribute alignment for successfully generated characters. Models with MLLM-based encoders or unified architectures demonstrate better alignment, evidenced by universally higher scores and a more gradual rate of decline with increasing prompt length, particularly for the "person" category.

Our benchmark provides evidence that employing MLLM-based encoders and developing unified model architectures are indeed promising directions for improving the performance of T2I models in long-prompt scenarios.

## U. Performance Robustness of Our Ablation Models on Short Prompts

Evaluating the performance of models fine-tuned for complexity on simpler inputs is crucial for validating their generalization capability. To discuss the performance of the fine-tuned models in our ablation study on short prompts, we conduct an evaluation using T2I-CompBench (Huang et al., 2023), a benchmark specifically designed for compositional T2I generation. The average prompt length in T2I-CompBench is significantly shorter, at only 12.65 tokens, making it an ideal evaluation setting for short prompts.

In this section, our evaluation rigorously assess prompt adherence across multiple dimensions, including: (1) Attribute Binding: color, shape, and texture; (2) Character Location: spatial relationships; (3) Character Interaction: non-spatial relationships; (4) Complex Composition: combination of multiple dimensions.

The evaluation results of the SDXL baseline and our four ablation models on T2I-CompBench are shown in Table 21:

*Table 21.* Performance robustness of our ablation models on T2I-CompBench.

| Model | Color | Shape | Texture | Spatial | Non-saptial | Complex |
|---|---|---|---|---|---|---|
| SDXL | 0.5710 | 0.4741 | 0.5176 | 0.2035 | 0.3105 | 0.3239 |
| 77 Limit w/ Short-Prompt-Training | 0.5701 | 0.4712 | 0.5184 | 0.1890 | 0.3079 | 0.3214 |
| 512 Limit w/ Short-Prompt-Training | 0.5862 | 0.4716 | 0.5168 | 0.1929 | 0.3074 | 0.3249 |
| 77 Limit w/ Long-Prompt-Training | 0.6057 | 0.4800 | 0.5417 | 0.2052 | 0.3105 | 0.3280 |
| 512 Limit w/ Long-Prompt-Training | 0.6108 | 0.4817 | 0.5608 | 0.2109 | 0.3112 | 0.3395 |

The results indicate that the performance of the fine-tuned models does not suffer negative degradation on short prompts.

- **Short-Prompt Trained Models:** Models trained exclusively on short prompts (*77 Limit w/ Short-Prompt-Training* and *512 Limit w/ Short-Prompt-Training*) show marginal fluctuations compared to the SDXL baseline, indicating that our short-prompt training process does not harm general compositional capability.

- **Long-Prompt Trained Models:** Crucially, the models trained using our detail-rich long prompts (*77 Limit w/ Long-Prompt-Training* and *512 Limit w/ Long-Prompt-Training*) exhibit all-around improvements. We observe significant boosts in the Attribute Binding metrics (especially Color and Texture), suggesting enhanced object-attribute association capability. The *512 Limit w/ Long-Prompt-Training* model also achieves the highest score in the Complex metric, indicating superior compositional fidelity even when dealing with concise prompts.

These results demonstrate that training on compositionally complex data successfully enhances the model's overall compositional capability, allowing it to adhere better to fine-grained constraints, even in the context of short prompts.

## V. Details on the Validation Process for High-Quality Prompts

Regarding the validation of our final prompts and annotations, as described in Section 3.1.2 and 3.1.3, we implement multiple measures to ensure data quality. These include: 1) deploying both the MLLM and an open-set object detection

model to identify main characters and exclude those with ambiguous localization; 2) in the character attribute extraction step, using the MLLM to verify whether each extracted attribute genuinely belongs to the main character (answering "yes" or "no"), and discarding mismatched attributes; and 3) during the prompt refinement and filtering phase, validating whether each annotation aligns with the final prompt, and discarding those that do not match. The specific details are presented below.

The first validation step in our pipeline focuses on achieving accurate and unambiguous character localization. To this end, we employ a dual-model verification approach, generating two distinct bounding box proposals for each identified character using both an MLLM and the open-set detector YOLOE-11L. We then reconcile these proposals based on their spatial agreement:

- If the two proposals have an Intersection over Union (IoU) of at least 0.7, indicating strong consensus, we select the larger one of the two boxes to ensure comprehensive coverage of the character. For instance, when tasked with localizing a "child wearing a gray hoodie," the proposals from YOLOE-11L ([121, 30, 357, 332]) and the MLLM ([135, 30, 364, 332]) yielded an IoU greater than 0.7. Therefore, we retained the larger YOLOE box.

- In cases where the IoU falls below 0.7, we employ BLIP to score the content in each box against the character's name. If the scores of the two boxes are both lower than a threshold of 0.4, we consider that both options are incorrect. Otherwise, the higher-scoring box is then retained. For example, in localizing a "person with a bag walking away," the YOLOE result ([278, 59, 365, 291]) and the MLLM result ([198, 57, 304, 302]) had a low IoU. However, the MLLM's proposal achieved a higher BLIP score and the score was more than 0.4. Hence, it was selected as the definitive location.

Our second validation step ensures the fidelity of the extracted character attributes. We iterate through every attribute associated with a character and use the MLLM to verify it with a strict verification prompt: "*Please analyze the main character in this image. Is 'temp_characteristic' one of its features? Only respond with 'yes' if it is a perfect match. Please only respond with 'yes' or 'no'.*" In this context, *temp_characteristic* represents the character attribute at the current iteration. If the MLLM responds with "no," we classify the attribute as a mismatch and discard it. This step is crucial for filtering out attributes that are ambiguous. For example, for a character described as a "child wearing a white shirt and blue shorts," the initially extracted attribute list was ['person', 'child', 'wearing blue jersey with number 12', 'short blonde hair', 'blue shorts', 'blue socks', 'playing soccer', 'white shirt']. During verification, the MLLM determined that the number '12' on the jersey was partially obscured, failing the "perfect match" criterion. As a result, the attribute "wearing blue jersey with number 12" was removed, yielding a more accurate, filtered list: ['person', 'child', 'short blonde hair', 'blue shorts', 'blue socks', 'playing soccer', 'white shirt'].

The third validation step takes place during the prompt refinement phase, guaranteeing that our final annotation data is perfectly synchronized with the content of the final prompts. We validate this alignment by instructing the MLLM to confirm whether each annotated attribute is explicitly described in the final prompt, using the guiding instruction: "*Your task is to determine whether the given image description includes the description of this particular CATEGORY feature of the main character.*" Here, *CATEGORY* denotes the current attribute category. Based on the MLLM's judgment, any attributes not found in the final prompt are removed from the annotation set. For example, for a character "beige labradoodle puppy sitting," the original attribute list was ['animal', 'labradoodle', 'puppy', 'sitting', 'fluffy', 'curly fur', 'soft expression']. After the MLLM's review of the final prompt, it was determined that the generic term "animal" and the subjective descriptor "soft expression" were not explicitly mentioned. The final, validated annotation list was therefore refined to ['labradoodle', 'puppy', 'sitting', 'fluffy', 'curly fur'].

# W. Case Studies

**'polished_prompt'**: "A high angle shot of a brown wooden bench with several dishes on top of it. In the center and on the left are two round, wavy side plates with black scratches on the sides and a doily pattern engraved on the plates. On both plates is a thick brown cookie that's been crosscut at the top, located in the middle part of the image. The plate on the right has a candy with a yellow wrapper and green ends. To the right of the plates is a white mug with whipped cream on top that is similar to the glass plates. The cup, made of ceramic material, has a cylindrical shape with a handle and a textured surface. The white whipped cream on top is frothy and has an embossed design. Surrounding the wooden bench is a dark brown wooden floor. On the top right is a gray curtain, and on the upper left is a view of the lower part of a white wooden wall. The image is taken indoors with soft, warm lighting, likely from an overhead source, creating a cozy and inviting atmosphere. The lighting is evenly distributed, with no harsh shadows, suggesting a relaxed time of day, possibly evening. The style of the image is a realistic photo with a warm, homely aesthetic. The brown wooden bench supports the two round, wavy side plates with black scratches and a doily pattern, which are placed side by side. The thick brown cookies crosscut at the top are positioned on top of the two round, wavy side plates, with one cookie on each plate. The candy with a yellow wrapper and green ends is located on the right plate, next to the thick brown cookie. The white mug with whipped cream on top is situated to the right of the two round, wavy side plates. The two round, wavy side plates are adjacent to each other, with the plate containing the candy being closer to the white mug with whipped cream on top."

**'character_attributes'**: [{'**main_character**': 'brown wooden bench', 'characteristics_list': ['several dishes on top'], 'cls': 'object'}, {'**main_character**': 'two round, wavy side plates with black scratches and a doily pattern', 'characteristics_list': ['two round, wavy side plates', 'black scratches on the sides', 'doily pattern engraved on the plates'], 'cls': 'object'}, {'**main_character**': 'thick brown cookies crosscut at the top', 'characteristics_list': ["thick brown cookie that's been crosscut at the top", 'thick brown cookie'], 'cls': 'object'}, {'**main_character**': 'white mug with whipped cream on top', 'characteristics_list': ['white mug', 'whipped cream', 'ceramic material', 'cylindrical shape', 'handle', 'textured surface', 'frothy topping', 'embossed design'], 'cls': 'object'}]

**'character_locations'**: [{'**main_character**': 'thick brown cookies crosscut at the top', 'bbox': [765, 740, 1098, 996], 'position': 'The middle part of the image'}]

**'scene_attributes'**: [{'scene_attribute': '**background**', 'content': 'The background features a dark brown wooden floor and a white wooden wall with a gray curtain on the right side.'}, {'scene_attribute': '**light**', 'content': 'The image is taken indoors with soft, warm lighting, likely from an overhead source, creating a cozy and inviting atmosphere. The lighting is evenly distributed, with no harsh shadows, suggesting a relaxed time of day, possibly evening.'}, {'scene_attribute': '**style**', 'content': 'The style of the image is a realistic photo with a warm, homely aesthetic.'}, {'scene_attribute': '**spatial**', 'content': ['The brown wooden bench supports the two round, wavy side plates with black scratches and a doily pattern, which are placed side by side.', 'The thick brown cookies crosscut at the top are positioned on top of the two round, wavy side plates, with one cookie on each plate.', 'The candy with a yellow wrapper and green ends is located on the right plate, next to the thick brown cookie.', 'The white mug with whipped cream on top is situated to the right of the two round, wavy side plates.', 'The two round, wavy side plates are adjacent to each other, with the plate containing the candy being closer to the white mug with whipped cream on top.']}]

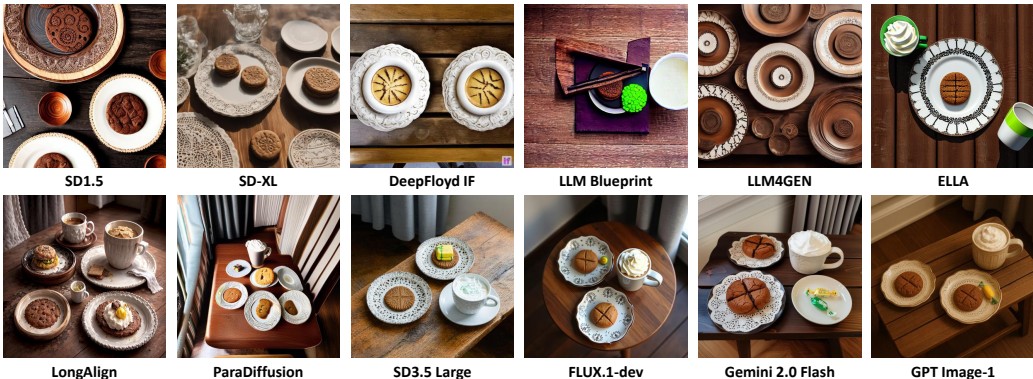

| SD1.5 | SD-XL | DeepFloyd IF | LLM Blueprint | LLM4GEN | ELLA |
| LongAlign | ParaDiffusion | SD3.5 Large | FLUX.1-dev | Gemini 2.0 Flash | GPT Image-1 |

*Figure 9.* Comparative visualization set for case study 1.

The first challenging case study reveals that long-prompt optimized models exhibit superior completeness in main character generation compared to their backbone counterparts, with the four most advanced models achieving the highest fidelity. Regarding fine-grained details such as "crosscut at the top", "cookies in the middle", and "a gray curtain on the right side", only the four most advanced models adhere to the instructions, with some remaining inaccuracies and omissions. While long-prompt optimized models (e.g., ParaDiffusion) interpret such details from the input, their generative capability remains constrained by backbone limitations, as evidenced by partial failures (e.g., generating cookies but omitting the crosscut).

**'polished_prompt'**: 'A medium-close-up view of a small shed that is made up of light brown wooden planks that run vertically, and along these planks there are two rectangular openings that are round along the top. Inside the windows, there are brown wooden planks that run vertically. In between these openings there is a blue banner that has two drawings of an owl in the middle and a woman to the left who is facing the owl, and the owl is standing on the woman\'s arm. To the left of the window, there is another banner that is light blue, and along it there is black text that reads "WILDLIFE REVE" vertically. The roof of the shed is triangular and gray and is being lit up by the sun, with the light source positioned to the front-left, casting soft shadows on the shed. In the upper part of the image, in front of the roof, there is a small falcon that is flying across and to the left, with its sharp beak, outstretched wings, streamlined body, and feathered tail, showcasing its natural predator instincts and agile flight. The falcon is positioned above the light blue banner with black text \'WILDLIFE REVE\'. Behind the shed is a tall tree that has a little bit of green foliage along it, and to the right of the tree are multiple trees that have no leaves along them. Behind the trees, a clear blue sky is visible, indicating a daytime setting in a natural or semi-natural environment. The image is brightly lit with natural sunlight, suggesting the light intensity is high, typical of midday. The style of the image is a realistic photo. It is daytime.'

**'character_attributes'**: [{'**main_character**': 'small shed with light brown wooden planks and two arched openings', 'characteristics_list': ['made up of light brown wooden planks', 'planks that run vertically', 'two rectangular openings that are round along the top', 'brown wooden planks that run vertically inside the windows', 'triangular and gray roof', 'roof being lit up by the sun'], 'cls': 'object'}, {'**main_character**': 'blue banner with drawings of an owl and a woman', 'characteristics_list': ['blue banner', 'two drawings of an owl', 'woman to the left', 'facing the owl', "owl is standing on the woman's arm"], 'cls': 'object'}, {'**main_character**': "light blue banner with black text 'WILDLIFE REVE'", 'characteristics_list': ['light blue banner', 'black text', 'WILDLIFE REVE'], 'cls': 'object'}, {'**main_character**': 'small falcon flying in front of the shed', 'characteristics_list': ['animal', 'small falcon', 'flying', 'sharp beak', 'outstretched wings', 'streamlined body', 'in motion', 'feathered tail', 'agile flight'], 'cls': 'animal'}]

**'character_locations'**: [{'**main_character**': 'small falcon flying in front of the shed', 'bbox': [674, 515, 906, 730], 'position': 'The upper part of the image'}]

**'scene_attributes'**: [{'scene_attribute': '**background**', 'content': 'The background features a clear blue sky and a mix of trees, some with green foliage and others bare, indicating a daytime setting in a natural or semi-natural environment.'}, {'scene_attribute': '**light**', 'content': 'The image is brightly lit with natural sunlight, indicating a daytime setting, and the light source is positioned to the front-left, casting soft shadows on the shed. The clear blue sky suggests the light intensity is high, typical of midday.'}, {'scene_attribute': '**style**', 'content': 'The style of the image is a realistic photo.'}, {'scene_attribute': '**spatial**', 'content': ["The small falcon flying in front of the shed is positioned above the light blue banner with black text 'WILDLIFE REVE'.", 'The blue banner with drawings of an owl and a woman is located between the two arched openings of the small shed with light brown wooden planks.', "The light blue banner with black text 'WILDLIFE REVE' is to the left of the blue banner with drawings of an owl and a woman.", 'The small shed with light brown wooden planks and two arched openings is situated behind the small falcon that is flying in front of it.']}]

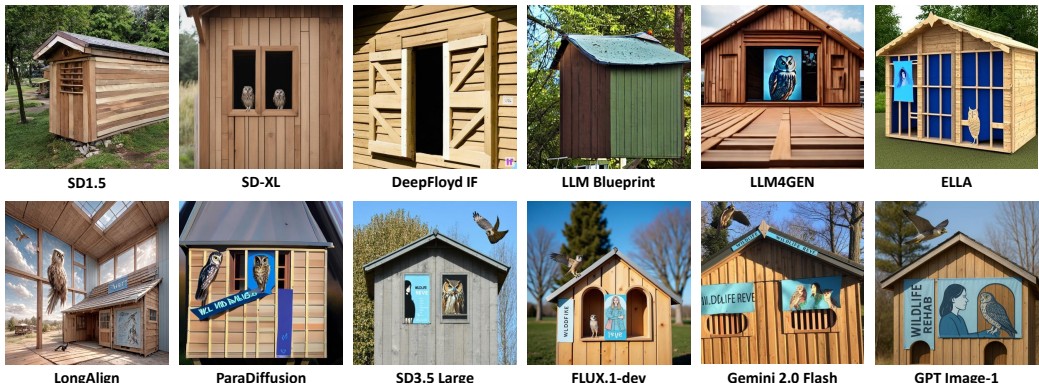

*Figure 10.* Comparative visualization set for case study 2.

The second challenging case study reveals systematic limitations across model categories: 1) Outdated models without long-prompt optimization fail to generate numerous specified characters, demonstrating catastrophic prompt adherence failures. 2) Long-prompt optimized models show improved yet incomplete character generation, indicating persistent architectural constraints. 3) Advanced models show critical shortcomings in detail fidelity. SD3.5 and FLUX fail to render the specified "blue banner with drawings of an owl and a woman", while Gemini 2.0 Flash and GPT Image-1 produce inaccurate textual elements ("WILDLIFE REVE"). These findings collectively underscore the ongoing challenges in compositional reasoning and detail preservation across current T2I models in long prompt scenarios.

