# OpenReview forum: "DetailMaster: Can Your Text-to-Image Model Handle Long Prompts?"
_ICML.cc/2026/Conference — ICML 2026 regular_

### Official Review · Reviewer_9F5c · 2026-03-01

**Soundness:** 3
**Presentation:** 3
**Significance:** 3
**Originality:** 3
**Overall Recommendation:** 4
**Confidence:** 4

**Summary:**

The paper introduces DetailMaster, a benchmark designed to evaluate the capability of Text-to-Image (T2I) models in following long, detailed prompts. While modern T2I models excel at short prompts, they suffer from significant performance degradation when constraints become dense and descriptions lengthen. To address this, the work proposes an automated data construction pipeline using MLLMs to generate high-quality image-text pairs and a multi-dimensional evaluation workflow covering Character Attributes, Locations, Scene Attributes, and Entity Relationships. The study aim to assess the concept of prompt adherence through an automated, MLLM-based evaluation protocol. The study benchmarks various SOTA models , revealing that training on long prompts is more critical than merely extending context windows.

**Compliance With Llm Reviewing Policy:**

Affirmed.

**Final Justification:**

Thanks for the thorough responses. The cross-evaluator validation with InternVL3 adequately addresses my concern about circular dependency, and the reframing of Scene Attributes as a baseline sanity check rather than a discriminative frontier is reasonable. The existing Mini-Benchmark and Lightweight Evaluator options also alleviate my concern about computational cost.

I will maintain my score.

**Key Questions For Authors:**

See the weaknesses.

**Limitations:**

Yes, the authors have adequately discussed the limitations.

**Strengths And Weaknesses:**

**Strengths:**

- Evaluating T2I models on complex and precise prompts is a critical and timely task for the current community as applications move toward professional, high-constraint generation.
- The DetailMaster benchmark fills a gap in evaluating model capabilities specifically for long-context, detail-rich generation, which has been under-explored compared to short-prompt alignment.
- The ablation studies offer pragmatic insights, effectively demonstrating that long-prompt training is more critical than merely extending context windows.

**Weaknesses:**

- The reliance on the Qwen model family for both data generation and evaluation creates a potential circular dependency. The benchmark might favor models with similar linguistic biases as the evaluator rather than objectively measuring adherence. Including a distinct, closed-source SOTA model (e.g., GPT-4o) as a judg for a subset would strengthen the claim of objectivity.
- The Scene Attributes metric appears to lack discriminative power for SOTA models, with recent models (e.g., FLUX, GPT Image-1) achieving near-perfect scores (>95%). This suggests that this specific dimension may not be challenging enough to distinguish future improvements.
- The evaluation workflow is computationally intensive, requiring an MLLM plus auxiliary tools for every generated sample. The paper lacks a detailed discussion on the cost/latency trade-off of running the full DetailMaster evaluation suite versus simpler alternatives.

---

> ### Author Rebuttal · Authors · 2026-03-31
>
> Dear Reviewer 9F5c, many thanks for your appreciation of the pragmatic insights of our work! We provide point-by-point responses to your valuable comments and sincerely hope that they can fully address your concerns.
>
> ---
>
> ## **W1: Evaluation Robustness and Bias Mitigation**
>
> Thank you for your feedback regarding the potential circular dependency in our evaluation pipeline. We would like to address this concern from the following two aspects:
>
>
>
> ### **1. Mitigation of Bias via Auxiliary Techniques and Cross-Evaluator Validation**
>
> Our evaluation avoids relying solely on MLLM priors by grounding judgments in actual image content via auxiliary techniques (e.g., open-set object detection, cropping, and structured verification). Additionally, we have already addressed the concern of "self-enhancement bias" in `Appendix J`. A control experiment evaluating Qwen-Image (which uses a Qwen2.5-VL encoder) with an architecturally distinct evaluator (InternVL3) yields consistently stable relative rankings. This proves DetailMaster measures genuine compositional fidelity, not evaluator-specific linguistic affinity.
>
>
>
> ### **2. Practical Considerations Regarding Closed-Source Models**
>
> Given our benchmark's massive scale—processing 74,931 multi-modal attributes across thousands of samples—using commercial APIs like GPT-4o incurs prohibitive costs that hinder broader community adoption. Instead, we deliberately utilize high-performance open-source MLLMs to ensure transparency, reproducibility, and accessibility. We have demonstrated the robustness of these evaluators through cross-evaluator consistency checks (see `Appendix J, K, and L`). Using these open-source evaluators, coupled with our auxiliary techniques, provides a more transparent and sustainable path for the community to evaluate long-prompt adherence.
>
>
> ---
>
>
> ## **W2: On the Discriminative Power of Scene Attributes**
>
> We would like to clarify the role of this metric in our benchmark:
>
> 1. **Analyzing Model Optimization for Classical Models:** The primary function of Scene Attributes is to evaluate holistic semantic fidelity and ensure that long-prompt optimization strategies do not compromise basic generation quality. Our experiments show that while SOTA models achieve near-perfect scores here, earlier models and some optimized variants show performance fluctuations.
> 2. **Highlighting Compositional Bottlenecks:** The high scores in Scene Attributes isolate the true frontier for future research. The saturation observed in this category shows that while advanced models excel at global scene composition, their primary bottlenecks remain in fine-grained attribute binding and spatial/interactive relationships. By confirming that scene fidelity is no longer a major hurdle for SOTA models, our benchmark naturally pivots the community's focus toward the more demanding and less solved dimensions of long-prompt generation.
>
>
> ---
>
>
> ## **W3: Justifying the Complexity of our Evaluation Pipeline**
>
> Thank you for your concern regarding the computational requirements of our evaluation workflow. We acknowledge that our pipeline is more intensive than traditional methods, but we argue this is a necessary trade-off to accurately measure fine-grained compositional fidelity:
>
> 1. **The Necessity of a Detailed Workflow for Compositional Accuracy:** Simpler benchmarks (e.g., DensePrompts and DPG-Bench) rely on CLIPScore or coarse VQA, which only assess global image-text alignment or basic existence. These methods are insufficient for the challenges of long-prompt generation. As seen in existing benchmarks such as T2I-CompBench, assessing the complex attributes inherently requires a multi-stage workflow involving object detectors and granular verification. Our workflow is designed to achieve this level of rigorous diagnostics. Without such a detailed pipeline, we would be unable to provide the fine-grained insights.
> 2. **Trade-off: Latency for High-Fidelity Diagnostics:** The latency observed in our workflow is proportional to the density of our annotations. By traversing every annotated attribute, our pipeline identifies where and why a model fails in long prompt scenarios. This level of scrutiny allows us to distinguish between models that truly follow a prompt versus those that rely on learned associations or coincidental alignment.
> 3. **Accessible Alternatives:** We are fully committed to accessibility for the research community. To address the computational burden, we have already provided two solutions in the paper: `Mini-Benchmark (Appendix N) and Lightweight Evaluator (Appendix K)`.
>
> ---
>
> ## **Closing Remarks**
>
> Thank you for the valuable time you have spent in reviewing our paper. We hope that our responses can address your concerns and strengthen your confidence in the acceptance of our paper.

---

> > ### Author Rebuttal · Reviewer_9F5c · 2026-04-05
> >
> > Thanks for the thorough responses. The cross-evaluator validation with InternVL3 adequately addresses my concern about circular dependency, and the reframing of Scene Attributes as a baseline sanity check rather than a discriminative frontier is reasonable. The existing Mini-Benchmark and Lightweight Evaluator options also alleviate my concern about computational cost.
> >
> > I will maintain my score.

---

### Official Review · Reviewer_HNek · 2026-03-03

**Soundness:** 3
**Presentation:** 3
**Significance:** 2
**Originality:** 2
**Overall Recommendation:** 5
**Confidence:** 4

**Summary:**

The paper simply contribute a benchmark for t2i generation models, which focus on the long-length input prompt, with considering various attributes, e.g., spatial relationship, or object properties.

**Compliance With Llm Reviewing Policy:**

Affirmed.

**Final Justification:**

The rebuttals address my concerns.

**Key Questions For Authors:**

See above. I prefer to accept the paper mainly because of the need for evaluating the t2i models with long-context input prompt.

**Limitations:**

Yes.

**Strengths And Weaknesses:**

The paper comprehensively evaluate the private and open-sourced models in the community on the proposed benchmark, showing the aspects for producing better generated images with long-context prompt.
I am happy about this benchmark because I think the community needs a benchmark for evaluating the t2i models under the situation with long-length input prompt.
The paper also introduces the details about pipeline for curating the benchmark.
I only have one question for the paper: is it possible to evaluate the nanobanana on the benchmark?

---

> ### Author Rebuttal · Authors · 2026-03-31
>
> Dear Reviewer HNek, thank you very much for your positive feedback! We are thrilled to hear your appreciation for the DetailMaster benchmark and our curation pipeline. We fully agree with your assessment that the community urgently needs robust, fine-grained benchmarks to evaluate T2I models under complex, long-context scenarios.
>
> ---
>
> To address your question regarding the evaluation of Nano Banana, we have newly supplemented our experiments with an evaluation of **Nano Banana 2** on our benchmark. Following the rigorous evaluation protocol detailed in our paper, we use Qwen2.5-VL-7B-Instruct as the sole MLLM evaluator and set the output resolution to 1024×1024 pixels. For context, we present its performance alongside the leading open-source and proprietary frontier models currently in our benchmark:
>
>
>
> | Model            | Object | Animal | Person | Character Locations | Background | Light | Style | Entity Relationships |
> | ---------------- | ------ | ------ | ------ | ------------------- | ---------- | ----- | ----- | -------------------- |
> | SD3.5 Large      | 48.56  | 46.20  | 32.95  | 33.62               | 89.61      | 90.33 | 95.69 | 40.03                |
> | FLUX.1-dev       | 51.47  | 45.83  | 34.91  | 41.57               | 95.77      | 97.05 | 94.81 | 47.49                |
> | Gemini 2.0 Flash | 55.44  | 47.84  | 34.23  | 44.74               | 96.69      | 95.90 | 97.20 | 50.78                |
> | GPT Image-1      | 59.41  | 48.04  | 40.40  | 53.92               | 97.50      | 98.85 | 97.69 | 63.07                |
> | Nano Banana 2    | 55.91  | 49.16  | 36.71  | 48.74               | 96.25      | 95.61 | 96.66 | 51.90                |
>
> Based on these results, we can draw several key insights regarding Nano Banana 2's capabilities in handling long and detail-rich prompts, which align with the findings in our paper (Line 294-303):
>
> 1. **Superiority Over Advanced Open-Source Models:**  Nano Banana 2 outperforms the advanced open-source models, such as SD3.5 Large and FLUX.1-dev, particularly in the structural bottleneck areas of "Character Locations" (48.74%) and "Entity Relationships" (51.90%). This confirms its advanced semantic extraction capacity for complex spatial and relational clauses.
> 2. **Competitive with Proprietary Frontier Models:** Nano Banana 2 operates at the same tier as the most advanced closed-source models. When compared to Gemini 2.0 Flash, Nano Banana 2 shows slight improvements across almost all fine-grained compositional dimensions, including Object (+0.47), Animal (+1.32), Person (+2.48), and Character Locations (+4.00).
> 3. **Performance Gaps and Future Directions:** Despite Nano Banana 2 being one of the latest proprietary T2I models (released after GPT Image-1), its performance in long-prompt scenarios still lags behind GPT Image-1. This suggests that **recent research may not have sufficiently prioritized long-prompt following optimization**, and the improved generation quality of new models might be largely confined to shorter prompts. Consequently, handling intricate, long-form instructions and mitigating "attribute leakage" remains an unsolved problem, leaving considerable room for improvement even for the most capable models in the community.
>
>
> We deeply appreciate your constructive suggestion, and we will definitely include the evaluation of this model in the final version of our paper!
>
> ---
>
> ## **Closing Remarks**
> We hope that our response has addressed your concern and can strengthen your confidence in the acceptance of our paper. Please do not hesitate to contact us if you have any further inquiries or recommendations. Thank you!

---

> > ### Author Rebuttal · Reviewer_HNek · 2026-03-31
> >
> > My main question is fully solved in rebuttal and I also have checked the other reviewers' comments and responses for them. I would love to maintain the current rating as A.

---

### Official Review · Reviewer_7zam · 2026-03-06

**Soundness:** 2
**Presentation:** 2
**Significance:** 1
**Originality:** 2
**Overall Recommendation:** 2
**Confidence:** 5

**Summary:**

This paper introduces **DETAILMASTER**, a benchmark designed to evaluate the ability of **text-to-image (T2I) models to follow long and complex prompts**. The authors argue that while modern T2I models perform well on short descriptions, they struggle with the long, structured prompts commonly required in professional applications. To address this gap, the paper proposes an automated data construction pipeline and evaluation workflow to build a benchmark consisting of **expert-validated prompts averaging ~285 tokens**. The benchmark evaluates models along four key dimensions: **Character Attributes, Structured Character Locations, Multi-Dimensional Scene Attributes, and Spatial/Interactive Relationships**. Experiments on a range of general-purpose and long-prompt-optimized T2I models show that existing systems still exhibit significant limitations in accurately binding attributes, handling compositional structures, and maintaining consistency under high constraint density, highlighting an important gap between current model capabilities and real-world long-prompt generation needs.

**Compliance With Llm Reviewing Policy:**

Affirmed.

**Final Justification:**

After reading the rebuttal and follow-up discussion, I appreciate the authors’ effort to include newer models and provide additional analysis. However, my main concerns remain. In my view, the paper’s core narrative and main-text conclusions are still built on a primary comparison that is not sufficiently representative of the current T2I landscape. I am also not fully convinced by some of the updated interpretations—for example, the conclusion drawn from the Nano Banana 2 results appears counter-intuitive and would require stronger analysis, and the claim that recent T2I development “neglects” long-prompt adherence seems overstated. More broadly, my concern is not about requiring controlled ablations, but that the paper is framed as a benchmark for text-to-image models in general, while the main evidence still centers on a relatively limited subset of models. Given this, I will maintain my score. Since I work closely on text-to-image evaluation and remain confident in this assessment after the rebuttal exchange, I will raise my confidence to 5.

**Key Questions For Authors:**

See the weakness part.

**Limitations:**

See the weakness part.

**Strengths And Weaknesses:**

### Strength
The paper focuses on an important problem of **long-prompt understanding in text-to-image generation**, which is highly relevant for practical use cases such as design, storytelling, and professional content creation.

### Weakness

1. As a benchmark paper submitted to a top-tier venue, it is important to provide a timely evaluation of recent models and a comprehensive comparison with existing benchmarks. However, the related work section has noticeable omissions. In the “text-to-image models” subsection, the most recent model discussed is GPT-Image (March 2025), which is already close to a year old at the time of submission. Many representative recent T2I models are missing, including the Qwen-Image series, Nano-Banana series, FLUX.2 series, Z-Image, among others. In addition, the discussion of model architectures mainly focuses on diffusion models, while other emerging paradigms such as autoregressive image generation models and unified multimodal models are not discussed. Similarly, in the “benchmarks for T2I generation” subsection, only benchmarks released in 2024 or earlier are covered. Several recent benchmarks from 2025, particularly those focusing on long prompt scenarios, are not discussed or compared, including LongBench-T2I [1], OneIG-Bench [2], and T2I-CoReBench [3].
2. Because the related work discussion is incomplete, some statements in the Introduction appear inaccurate or at least outdated. For example, the paper claims that *“only a few benchmarks consider long prompts”* and that existing works *“often rely on coarse metrics (e.g., CLIPScore)”*. Given the recent developments mentioned above, these statements may no longer hold, which weakens the motivation of the paper.
3. Due to the limited discussion of recent T2I models, the evaluation in Table 2 mainly includes relatively older systems. As a result, some of the conclusions drawn from these results may have limited practical relevance. For example, in Section 4.2.2, the discussion of *optimized methods* focuses on approaches that improve the T5 text encoder. However, many recent SOTA T2I models no longer rely on T5, and instead adopt LLM- or MLLM-based text encoders. Therefore, the insights derived from these comparisons may not reflect the current state of the field. Similarly, the statement *“indicating that even SOTA models have considerable room for improvement in long prompt scenarios”* refers to GPT-Image as the SOTA model, which is no longer the case compared to more recent systems such as Nano Banana Pro.
4. The appendix includes evaluations of several newer diffusion and unified models, which I appreciate. However, placing these results only in the appendix seems inappropriate for a benchmark paper. Ideally, all major evaluation results should appear in the main comparison table, as the conclusions in the main text are currently derived from a set of outdated baseline models, reducing the significance of the analysis.
5. The benchmark construction and evaluation rely heavily on Qwen2.5-series LLM/MLLM models. While I understand that updating the entire dataset generation pipeline may be difficult once the prompts have already been constructed, the evaluation stage still uses Qwen2.5-VL-7B-Instruct as the sole evaluator. This choice appears somewhat outdated given the rapid progress in multimodal models (compared to Qwen3 and Qwen3.5 series). The limited capability of this model may introduce non-negligible evaluation errors, especially for complex long-prompt reasoning. Moreover, the paper does not provide a human alignment study or a clear justification for selecting this particular model as the evaluator.



[1] Draw all your imagine: A holistic benchmark and agent framework for complex instruction-based image generation

[2] Oneig-bench: Omni-dimensional nuanced evaluation for image generation, NeurIPS'25

[3] Easier Painting Than Thinking: Can Text-to-Image Models Set the Stage, but Not Direct the Play?, ICLR'26

---

> ### Author Rebuttal · Authors · 2026-03-31
>
> Dear Reviewer 7zam, we sincerely appreciate the time and effort you have dedicated to reviewing our paper! Below, we provide detailed responses to each of your comments, with the hope that this will encourage you to lean toward the acceptance of our paper.
>
> ---
>
> ## **Part 1: Timeliness**
>
> ### **1.1: *"... recent models are missing."* (W1 & W2 & W4)**
>
> Thank you for your suggestion to include more recent T2I models. In fact, we have already conducted evaluations on advanced models —including **Qwen-Image, Janus-Pro, Lumina-Image-2.0, and BAGEL**—which were detailed in `Appendix O and Appendix P`. We have also followed your suggestion to introduce more advanced T2I models, such as **Nano Banana 2** (see **1.3**).
>
>
> ### **1.2: *"Given the recent developments, these statements may no longer hold ..."* (W1 & W3 & W4)**
>
> Our focus on classical models was a deliberate design choice. Keeping base architectures constant allows for controlled ablations and alignment with established benchmarking practices, isolating the impact of context windows, training strategies and architectures. The insights drawn from these controlled studies are applicable to various T2I frameworks.
>
> Nevertheless, we agree that advanced models represent the current frontier. We will leverage the extra one page allowed in camera-ready version to integrate their results into the main text.
>
>
>
> ### **1.3: *"Newer models have surpassed GPT-Image, making SOTA performance claims wrong."* (W1 & W3)**
>
> We have newly evaluated the SOTA model Nano Banana 2 (`see our response to Reviewer HNek`). The results show that even the most advanced models still have room for improvement in following long prompts.
>
>
> ---
>
>
> ## **Part 2: Comparison with More Benchmarks (W1 & W2)**
>
> Thank you for highlighting these excellent works! We will add a dedicated discussion to compare them in our final version. However, while they share the goal of evaluating complex prompts, DetailMaster maintains distinct and valuable advantages:
>
> **1. Prompt Source & Visual Logic**
>
> - LongBench-T2I & T2I-CoReBench: Rely heavily on LLM-generated/synthetic prompts, which risk violating real-world physical logic.
> - DetailMaster: Grounded in real image-caption pairs, ensuring visual realism, structural logic, and semantic diversity.
>
>  **2. Density of Long Prompts**
>
> - OneIG-Bench: Only ~25% of prompts exceed 60 tokens.
> - DetailMaster: Avg. 284.9 tokens.
>
>  **3. Evaluation Granularity & Rigor**
>
> - OneIG-Bench & T2I-CoReBench: Rely on coarse overall semantic alignment or basic VQA checklists.
> - DetailMaster: Employs a hierarchical, multi-model verification pipeline. We evaluate fine-grained adherence—isolating individual attributes, locations, and relationships.
>
> Therefore, DetailMaster remains a uniquely robust tool for researchers targeting fine-grained prompt adherence in real-world long prompt scenarios.
>
> ---
>
> ## **Part 3: Rationale of the Evaluation Pipeline**
>
> ### **3.1: Justification against Qwen3/3.5 (W5)**
>
> We opted against Qwen3/3.5 because we previously found a critical misalignment with human annotators: it tends to overly penalize low-aesthetic images (e.g., SD1.5 outputs), marking successfully generated but poorly rendered objects as "failed." Qwen2.5-VL avoids this bias, providing fair semantic evaluations. Furthermore, as already detailed in `Appendix H`, Qwen2.5-VL shows the highest correlation with human judgment among the tested alternatives.
>
>
>
> ### **3.2: New Evaluation with Qwen3.5 (W5)**
>
> We have newly re-evaluated all models in Table 2 using Qwen3.5-9B. Due to character limits, we display the table in the anonymous link: https://anonymous.4open.science/r/DetailMaster-6DE8/evaluation_with_qwen3_5.pdf
>
> From the results, we can see that:
>
> 1. SD1.5 backbone scores drop noticeably, confirming Qwen3.5 rejects low-quality but semantically correct generations.
> 2. The Kendall's Tau Correlation Coefficients across metrics between Qwen3.5 and Qwen2.5-VL (0.88, 0.85, 0.91, 1.0, 1.0, 0.85, 0.91, 0.97) are high and reflect a strong positive correlation, providing evidence that our benchmark's conclusions are robust and not an artifact of the specific MLLM used.
>
>
>
> ### **3.3: Tool-Augmented Pipeline & Evaluator Bias (W5)**
>
> Our evaluation does not solely rely on the MLLM. Auxiliary tools handle spatial grounding and feature isolation, leaving the MLLM to make localized "yes/no" decisions. Furthermore, `Appendix J` validates our pipeline using InternVL3, which preserves model rankings and proves no self-enhancement bias (Qwen-Image's strong performance holds under non-Qwen evaluators).
>
> ---
>
> ## **Closing Remarks**
>
> We want to express our deepest gratitude for your constructive suggestions. We will leverage the extra one page allowed in camera-ready version to incorporate your recommendations. We hope that you can consider an increase in the rating if these responses with new results and planed modifications address your comments. Thank you!

---

> > ### Author Rebuttal · Reviewer_7zam · 2026-04-03
> >
> > Thank you for the rebuttal and the additional appendix results. After reading the paper more carefully, I agree that the submission does include some evaluations of newer model families in the appendix. However, my main concern remains that the paper’s core narrative, primary comparison table, and main-text conclusions are still built mostly on the model set in Table 2, which is not sufficiently representative of the current state of the field. While the appendix expands the model coverage, the main-text conclusions are still derived primarily from Table 2 and therefore do not adequately reflect recent developments in T2I models. As a benchmark paper, this is a substantial issue, since newer architectures and more recent related benchmarks are not adequately integrated into the main text.
> >
> >
> > I am also not convinced that restricting the main comparison to a narrower set of “classical models” makes the analysis sufficiently fair or controlled. Even within a similar architectural family, the evaluated models still differ substantially in scale, training data, training objectives, and inference pipelines, so it is difficult to attribute the observed differences cleanly to any single factor. In addition, the paper is framed as an evaluation of Text-to-Image models in general, while the main-text evidence is still centered on a relatively limited subset of that broader landscape (e.g., diffusion models).
> >
> > Overall, my concern is not that the paper lacks useful evidence, but that the current main-text evidence is not strong or timely enough to support the breadth of its claims. Addressing this would require a substantial revision of the main comparison, related-work positioning, and conclusion framing, which I do not think can be fully resolved within the scope of the current rebuttal. Therefore, I will maintain my current score.

---

> > > ### Author Response · Authors · 2026-04-04
> > >
> > > Dear Reviewer 7zam,
> > >
> > > Thank you for your follow-up comments and for reviewing our appendix evaluations!
> > >
> > > We respectfully clarify that **the necessary elements for the requested enhancements were already in our submission** (Sections 4.2, 4.4, and Conclusion), where we discussed advanced models, LLM/MLLM-based encoders, and unified models. Enhancing the paper involves: (1) updating Table 2 and Figure 3 with appendix table/figure, (2) updating Related Works/Intro on recent benchmark differences, (3) further validating evaluator validity with Qwen3.5, and (4) using the camera-ready extra page to expand our existing discussions.
> > >
> > > Below, we clarify our concrete revisions and address your remaining points.
> > >
> > > ---
> > >
> > > ## **1. Revision of Table 2 and Expansion of Main-Text Analysis**
> > >
> > > Following your suggestion, we have **newly updated Table 2 to include more advanced models**. We incorporate **the new SOTA Nano Banana 2, models with superior LLM/MLLM encoders, and unified models**, which are predominantly from 2025/2026. The updated Table 2 is displayed at the anonymous link: https://anonymous.4open.science/r/DetailMaster-6DE8/rebuttal_new_table2.pdf
> > >
> > > Crucially, integrating these models **reinforces our existing main-text conclusions** rather than altering them:
> > >
> > > - **Performance Ceiling of SOTA Models: Nano Banana 2** shows results close to Gemini 2.0 Flash, indicating room for improvement in long-prompt scenarios. Notably, despite being more recent than GPT Image-1, Nano Banana 2 scores lower. **This strengthens our insight: T2I model development still neglects long-prompt adherence;** recent gains are often confined to image aesthetics under shorter prompts.
> > > - **Deepening Section 4.2.2:** We expand the existing paragraph on "Advanced LLM/MLLM encoders and unified architectures" into two targeted analyses.
> > >   - *Advanced LLM/MLLM Encoder:* A stronger encoder improves semantic extraction, yet even these models lag on the hardest dimensions (Locations and Relationships), showing that better encoders alone are insufficient.
> > >   - *Unified Models:* Specific architectural choices (e.g., decoupled encoders or MoT-style designs), richer training data (e.g., UniCap-style scaling), and larger model scale benefit performance under long prompts.
> > > - **Error Patterns Remain Consistent:** We have newly examined the error cases for recent models such as Qwen-Image and BAGEL-7B-MoT. **Our original error source analysis still perfectly apply:** 1) conflicts with real-world positional priors; 2) difficulty in rendering complex interactions/apparel; 3) cascading failures from attribute errors to relationship errors; 4) failures in structural complexity.
> > > - **Validation of Trends (Fig. 3):** As shown in Appendix Fig. 7 and 8, the negative correlation between generation accuracy and prompt token length, as well as the fact that stronger models mitigate attribute misalignment, hold fully for recent models. Merging these into the main text strengthens Section 4.3 without requiring rewrite.
> > >
> > > In summary, the addition of the new SOTA model and the integration of recent models from our appendix support and reinforce our existing main-text conclusions. The required updates are structural expansions, not a substantial revision.
> > >
> > > ---
> > >
> > > ## **2. Recent Benchmarks and the "Fair Comparison" Argument**
> > >
> > > We respectfully clarify that if we apply the standard—that comparing models is not a strictly "controlled ablation" because training data and inference pipelines differ—**then benchmarks such as OneIG-Bench face the same challenge** when discussing the advantages of SD1.5’s data cleaning, Recraft V3’s layout strategy, or T5-based models. In evaluating advanced models, differences in training data are unavoidable. Benchmarking practice thus attributes major performance shifts to the primary innovation of contemporaneous models.
> > >
> > > DetailMaster goes a step further. Reviewers H4TM, HNek, and 9F5c reached a consensus and praised our benchmark for "comprehensively evaluating" a wide spectrum of models and yielding "pragmatic insights" through well-designed ablations. By comparing models that share backbones but employ distinct optimizations (**controlling the backbone variable better than other benchmarks**), and **conducting controlled ablation studies in Section 4.5 and Table 3**, DetailMaster is able to draw specific optimization insights—such as longer training prompts aid complex interpretation.
> > >
> > > ---
> > >
> > > ## **Closing Remark**
> > >
> > > We sincerely hope that these concrete, evidence-backed updates address your concerns and encourage you to re-evaluate our paper. Thank you again for your valuable feedback.

---

### Official Review · Reviewer_H4TM · 2026-03-09

**Soundness:** 3
**Presentation:** 3
**Significance:** 3
**Originality:** 3
**Overall Recommendation:** 4
**Confidence:** 3

**Summary:**

This paper introduces DETAILMASTER, a comprehensive benchmark for evaluating text-to-image (T2I) generation models on long, compositionally complex prompts averaging 284.89 tokens. The benchmark comprises 4,116 expert-validated prompts with fine-grained annotations across four dimensions: Character Attributes, Structured Character Locations, Multi-Dimensional Scene Attributes, and Spatial/Interactive Relationships. The authors develop an automated data construction pipeline leveraging MLLMs and auxiliary techniques to extract and augment attributes from existing caption datasets (DOCCI, Localized Narratives). They propose a robust evaluation workflow using MLLM-based assessment with structured verification steps. Extensive experiments on 12 models reveal that even state-of-the-art systems struggle with long prompts, showing degraded adherence as prompt length increases.

**Compliance With Llm Reviewing Policy:**

Affirmed.

**Final Justification:**

My concerns have been adequately addressed.

**Key Questions For Authors:**

1. Have you considered conducting even a small human study to directly assess the generated images? This would validate whether your automated scores align with human perception of quality and compositional correctness.

2. MLLMs also struggle with spatial reasoning—the very task you identify as hardest for T2I models. What is the error rate of your Qwen2.5-VL evaluator on these challenging tasks? Can you quantify how often the evaluator itself makes mistakes, potentially distorting the reported model scores?

**Limitations:**

See Weaknesses

**Strengths And Weaknesses:**

Strengths

1. Well-Motivated Problem and Comprehensive Benchmark Design.
2. Extensive Empirical Evaluation: The paper evaluates 12+ models spanning proprietary systems (GPT Image-1, Gemini 2.0 Flash), general-purpose open-source models (SD1.5, SD-XL, FLUX, SD3.5), and long-prompt-optimized variants, with additional evaluations in appendices. The controlled ablation study systematically disentangles token limits from training data length, yielding clear insights about what drives performance.
3. Strong Robustness Validation: The submission strives to assess the concept of evaluation robustness through multiple checks. Appendix J demonstrates consistent model rankings when switching evaluators (Qwen2.5-VL-7B to InternVL3-9B),


Weaknesses

1. The paper did not conduct a human quality assessment of the model's generated results （a small set）, which raises concerns about the fairness and reasonableness of the metrics.

2. Artificial Nature of Long Prompts: The automatically generated prompts, averaging over 280 tokens, often feel synthetic and redundant due to templated expansion. It is unclear how well these "machine-generated" stress-test prompts represent the natural distribution and semantic coherence of long-form descriptions from real users. The paper lacks discussion on the real-world applicability of its findings.

3. Unquantified Evaluator Fallibility: The evaluation heavily relies on an MLLM evaluator, yet MLLMs themselves struggle with the same fine-grained spatial and relational reasoning tasks that this benchmark targets. The paper fails to quantify the potential error bounds of its MLLM-based assessment, especially when the evaluator might misjudge complex relationships, casting uncertainty on the reported accuracy scores.

---

> ### Author Rebuttal · Authors · 2026-03-31
>
> Dear Reviewer H4TM, we sincerely appreciate your positive recognition of the extensive empirical evaluation and the strong robustness validation in our work! In response to your comments, we provide detailed discussions along with additional experiments below. We hope our responses can fully address your concerns.
>
> ---
>
> ## **W1 & Q1: Human Evaluation**
>
> Thank you for this constructive suggestion. While we previously performed human verification to ensure data quality (see `Appendix G`), we acknowledge that an additional human assessment of the model-generated images helps validate the correlation between our automated evaluation pipeline and human perception.
>
> To address your concern, we have newly conducted a small-scale human quality assessment. We randomly select 50 samples, encompassing 441 Object Attributes and 75 Character Locations, and perform a comparative evaluation with three models:
>
> | Model       | Human Assessment: Object Attributes | Human Assessment: Character Locations | Our Pipeline: Object Attributes | Our Pipeline: Character Locations |
> | ----------- | ----------------------------------- | ------------------------------------- | ------------------------------- | --------------------------------- |
> | SD-XL       | 27.7                                | 12.0                                 | 23.8                            | 6.7                               |
> | SD3.5 Large | 50.6                                | 36.0                                  | 48.5                            | 33.3                              |
> | GPT Image-1 | 57.8                                | 58.7                                  | 53.5                            | 58.7                              |
>
> From the results, there are two findings:
>
> 1. **Robustness:** There is a strong correlation between the scores derived from human assessment and our pipeline, providing empirical evidence that our metrics are reasonable and robust.
> 2. **Human Lenience:** The human-assessed scores are slightly higher than those obtained from our pipeline (averaging +3.4 and +2.7). This discrepancy may originate from the fact that human evaluators are prone to cognitive association, often mentally "filling in" distorted details when image quality is poor.
>
> In summary, our automated pipeline provides a balance between high-fidelity assessment and scalability.
>
> ---
>
>
> ## **W2: Stress-Testing vs. Realistic Distribution**
>
> We appreciate the opportunity to clarify that DetailMaster is positioned as a stress-test benchmark rather than a representative sample of general user query distributions.
>
> The primary objective of our automated prompt expansion is to simulate "detail-overload" scenarios, purposefully creating high constraint density, multi-entity relationships, and positional requirements. This design is intended to probe the failure boundaries of T2I models under complex compositional pressure.
>
> While casual user prompts are often shorter, there are demanding requirements of professional creative workflows (e.g., high-fidelity advertising design or cinematic storyboarding). These applications require sophisticated instructions to realize a specific vision. Furthermore, modern AIGC pipelines increasingly rely on automated prompt expansion—such as DALL-E 3's caption upsampling—to enrich simple user inputs. By making these implicit complexities explicit, DetailMaster serves as a proxy for professional-grade prompt adherence.
>
> In our final version, we will clearly distinguish between a "realistic prompt distribution benchmark" and our "detail-overload stress benchmark".
>
>
> ---
>
>
> ## **W3 & Q2: Evaluator Robustness**
>
> We would like to clarify that our evaluation is not a "pure" MLLM assessment. It is a hybrid framework that reinforces MLLM with auxiliary tools to ground the evaluation in objective visual features.
>
> To address your specific points:
>
> 1. **Auxiliary Grounding:** In `Appendix H.3`, we analyzed the impact of our auxiliary techniques:
>
> - - By integrating an object detector, we automatically corrected 16.6% of bounding box coordinates that were misaligned by the MLLM alone.
>  - - Our ablation studies show that omitting image cropping leads to a 15.1% reduction in attribute extraction, proving that this approach reliably captures details that are difficult for MLLMs.
>
> 2. **Robustness of Relative Model Rankings:** While calculating an absolute "error rate" is difficult in the absence of a perfect evaluator, we ensure benchmark validity through consistent relative benchmarking. As shown in `Appendix J`, re-evaluating with a different MLLM (i.e., InternVL3) preserves the relative performance rankings across models. In addition, as shown in our response to W1 & Q1, our automated results correlate with human assessments.
>
> ---
>
> ## **Closing Remarks**
>
> We sincerely appreciate the valuable time you have spent in reviewing our paper. Please do not hesitate to contact us if you have any further inquiries or recommendations.

---

> > ### Author Rebuttal · Reviewer_H4TM · 2026-04-03
> >
> > Thank you for the thorough rebuttal.

---

### Decision · Program_Chairs · 2026-04-30

**Decision:**

Accept (regular)

**Comment:**

The reviewers agreed that this paper addresses an important and timely problem by introducing a benchmark for evaluating text-to-image models on long, compositional prompts, and they found the benchmark design, breadth of evaluation, and practical insights generally valuable. Before rebuttal, the main concerns centered on the lack of human or cross-evaluator validation, possible evaluator bias and cost, the scope and clarity of the comparisons, and whether the benchmark’s model coverage and main-text framing were sufficiently representative of the rapidly evolving T2I landscape. After rebuttal, three reviewers concluded that their concerns were adequately addressed or substantially clarified, citing the added validation, stronger discussion of evaluation robustness, and expanded analysis with newer models, and they maintained positive recommendations. One reviewer remained unconvinced, arguing that the paper’s main narrative still relies too heavily on a limited comparison set and that some updated interpretations require stronger support; however, this concern appears to be more about framing and representativeness than a fundamental flaw in the benchmark itself. Overall, given the clear positive consensus after rebuttal and the paper’s solid technical contribution, AC recommends accept.